# Distinct manifestations of excitatory-inhibitory imbalance associated with amyloid-β and tau in patients with Alzheimer's disease

Kamalini G. Ranasinghe [1] ✉, Kiwamu Kudo [2], Faatimah Syed[1], Claire Yballa[1], Joel H. Kramer[1], Bruce L. Miller[1], Katherine P. Rankin [1], Paul A. Garcia[3], Heidi E. Kirsch [3], Keith Vossel[1,4], William Jagust [5], Gil D. Rabinovici [1,2] & Srikantan S. Nagarajan[2]

A growing body of evidence shows that epileptic activity is frequently observed in patients with Alzheimer's disease (AD), implicating underlying excitatory-inhibitory imbalance. The distinction of whether the AD-epileptic phenotype represents a subset of patients or an underdiagnosed manifestation holds major therapeutic implications. Here, we quantified the excitatory-inhibitory imbalance in AD patients using magnetoencephalography and examined the relationships to AD pathophysiology—amyloid-beta and tau, and to epileptic activity. We used two metrics to quantify regional excitatory-inhibitory imbalance distinguishing between local hyperexcitability (*Neural excitability, quantified by regional aperiodic spectral slope*) and aberrant long-range synaptic input integration (*Neural fragility, quantified by regional linear dynamic instability*). We found that amyloid-beta correlated with higher neural fragility and higher neural excitability, while tau and hypometabolism uniquely correlated with higher neural excitability. Importantly, the AD-epileptic phenotype showed a distinctive increase in neural fragility. Our findings demonstrate that AD pathophysiology is associated with diverse mechanisms of excitatory-inhibitory imbalance and that AD-epileptic phenotype represents a distinct group of patients with greater impairments in long-range synaptic input integration.

Overwhelming evidence supports a model of Alzheimer's disease (AD) pathogenesis requiring both amyloid-beta (Aβ) and tau accumulation to provoke substantial cognitive decline. Aβ accumulation occurs early and involves much of the neocortex by the time the earliest neocortical tau is detected, subsequently followed by cognitive decline[1]. Several recent fundamental discoveries have highlighted that excitatory-inhibitory (E/I) imbalance, among other critical cellular processes, plays an important intermediate pathogenic role in the early phase of

[1]Memory and Aging Center, Department of Neurology, University of California San Francisco, San Francisco, CA, USA. [2]Department of Radiology and Biomedical Imaging, University of California San Francisco, San Francisco, CA, USA. [3]Epilepsy Center, Department of Neurology, University of California San Francisco, San Francisco, CA, USA. [4]Mary S. Easton Center for Alzheimer's Disease Research, Department of Neurology, David Geffen School of Medicine, University of California Los Angeles, Los Angeles, CA, USA. [5]Department of Neuroscience, UC Berkeley, Berkeley, CA, USA.
✉e-mail: Kamalini.ranasinghe@ucsf.edu

AD, between Aβ accumulation and clinical disease[2]. In AD mouse models, pathological forms of both Aβ and tau have been documented to alter E/I activity of neural circuits. For example, in Aβ overexpressing mice, human amyloid precursor protein (hAPP) and its metabolites are implicated in increased firing of excitatory pyramidal neurons by suppressing glutamate reuptake, as well as impaired firing of inhibitory neurons by depleting their voltage-gated sodium channel subunit Nav1.1[3–5]. Likewise, in tau deficient knock-in mice ($MAPT^{-/-}$), reducing endogenous wild-type tau has been shown to increase the activity of inhibitory neurons, suggesting tau associated imbalance of E/I activity[6–8]. However, despite significant progress in the development of AD mouse models[9,10], none capture the full complexity of AD pathobiology and hence fall short of characterizing the relationship between E/I imbalance and AD pathophysiology in its entirety.

Consistent with the phenomenon of altered E/I balance, a growing body of work shows that an estimated 20–60% (depending on the sensitivity of the method) of AD patients have epileptic manifestations[11–15]. Importantly, these studies demonstrate that epileptic abnormalities manifest early in the disease course, often precede the onset of cognitive decline, and have high incidence in both sporadic and in autosomal dominant AD[16–19]. Furthermore, AD patients who harbor interictal spikes and sharp waves progress faster than those who were negative for such abnormalities–an indication that altered E/I balance is allied to the clinicopathological presentation[10,18]. Collectively, both basic and clinical science evidence suggest that E/I imbalance in AD represents a central pathogenic mechanism. However, this raises the intriguing question of why epileptic manifestations are not universally observed in all AD patients. One potential explanation is that while E/I imbalance in AD consistently contributes to epileptic abnormalities, these manifestations may only be detected in a subset of patients due to the limited sensitivity of electrophysiological assays. Alternatively, E/I imbalance in AD may involve diverse mechanisms–some leading to neuronal dysfunction present in all patients, and others specifically contributing to epileptic activity in a subset. Notably, findings in AD transgenic mouse models support such diversity. For instance, Aβ-overexpressing mice consistently exhibit abnormal hyperactivity of neurons including seizures and interictal spikes that correlate with Aβ burden[5,20,21]. In contrast, tau transgenic mice present mixed outcomes, with both hyper- and hypo-activity of neurons[6,22,23]. These results indicate that Aβ and tau may contribute differentially to altered E/I balance in AD mouse models. However, it remains unknown in AD patients whether and to what extent Aβ and tau are implicated in differential manifestations of E/I imbalance.

## Table 1 | Participant demographics

|  | AD (n = 82) | Controls (n = 40) | p-value |
|---|---|---|---|
| Age – yr | 62.3 ± 8.9 | 64.0 ± 5.2 | 0.07 |
| Female sex – no. (%) | 45 (54.9) | 24 (60.0) | 0.59 |
| Right handedness – no. (%) | 71 (86.6) | 31 (79.5) | 0.32 |
| White Race – no. (%)[a] | 78 (96.29) | 33 (86.8) | 0.17 |
| Education – yr | 16.5 ± 2.7 | 17.5 ± 1.7 | 0.01 |
| MMSE[b] | 22.0 ± 5.2 | 29.6 ± 0.6 | <0.0001 |
| CDR[c] | 0.7 ± 0.2 | 0 ± 0 | <0.0001 |

Values for age, education, MMSE, and CDR indicate means ± standard deviations.
Statistical comparisons of sex, handedness, and race was performed using Chi-Square test.
Statistical comparisons of age, education, MMSE, and CDR were performed using Wilcoxon Mann-Whitney test.
*AD* Alzheimer's disease, *CDR* clinical dementia rating, *MMSE* mini mental state exam.
[a]Race was self-reported, and the total number of observations reported for each group included n = 81, and n = 38 for patients with AD and control groups, respectively.
[b]Scores on the Mini Mental State Exam (MMSE) range from 0 to 30, with higher scores denoting better cognitive function.
[c]Scores on the CDR range from 0 to 3 with higher scores denoting greater impairment.

In this study, we investigated the associations between AD pathophysiology and regional E/I imbalance, using two complementary metrics. First, we quantified regional E/I imbalance due to local hyperexcitability arising from intrinsic neuronal and synaptic processes, which is reflected in aperiodic activity of the neural power spectrum–termed *Neural excitability*[24,25]. Second, we quantified abnormal long-range synaptic input integration, using a *Neural fragility*[26,27] metric that estimates the linear dynamic instability within a local neural ensemble. The study included high spatiotemporal resolution magnetoencephalography (MEG) imaging in cohorts of well characterized, biomarker positive early-stage AD patients (n = 82) and age-matched controls (n = 40). Subsets of AD patients were evaluated with multimodal imaging combining MEG and positron emission tomography (PET) with tracers to quantify Aβ ($^{11}$C-PIB), tau (flortaucipir), and hypometabolism ($^{18}$F-FDG), and with long-term electroencephalography (LTM-EEG) with video monitoring combined with 1-hour simultaneous MEG and EEG (M/EEG) to identify epileptic activity (AD-EPI+ vs AD-EPI −). We tested the hypotheses that Aβ, tau, and FDG uptakes have distinct associations with neural excitability and neural fragility, and that AD-EPI+ will represent a distinct phenotype of E/I imbalance in AD.

## Results

### Participants, neural fragility, and neural excitability

We used high spatiotemporal MEG imaging to record resting-state neural activity patterns in a cohort of 82 biomarker positive AD patients, and 40 age-matched controls, recruited from the Memory and Aging Center, University of California San Francisco (Table 1). Patients with AD belonged to the early biological stages of AD reporting either 0.5 (n = 45) or 1 (n = 37) on the clinical dementia rating (CDR) at the time of MEG. In patients with AD and age-matched controls, we quantified regional E/I imbalance (210 cortical regions, Brainnetome atlas)[28] using two complementary metrics. First, *Neural excitability* (estimated as aperiodic spectral slope), which quantified E/I imbalance resulting from local hyperexcitability due to aberrant intrinsic neuronal and synaptic processes. Specifically, using the source localized MEG signal, we estimated the spectral aperiodic slope in the regional power spectrum of 15–50 Hz frequency band, which captures the non-oscillatory, intrinsic activity of local neuronal ensembles[29] (Fig. 1). Flatter aperiodic slopes (higher numerical values) have been shown to indicate local hyperexcitability of neurons with reduced inhibitory regulation[25,30,31]. Second, *Neural fragility* (estimated as linear dynamic instability), which quantifies E/I imbalance triggered from abnormal long range synaptic inputs. Neural fragility[26,27,32] is a quantitative measure of linear dynamic instability within a local neuronal ensemble and estimated as the minimum energy perturbation required to alter E/I balance when integrating long-range synaptic inputs. The greater the fragility of a region, the higher the probability to alter E/I balance with a minimum energy perturbation. Specifically, we used the source localized MEG timeseries and estimated the minimum required energy (row perturbation)[27] to destabilize input-integration from the whole brain network (Fig. 1). These two metrics, although conceptually distinct, capture complementary aspects of E/I dynamics. Importantly, neural excitability primarily reflects intrinsic local activity that in turn also incorporates contributions from all synaptic inputs. In contrast, neural fragility explicitly models the impact of long-range synaptic input integration on E/I balance.

### Distinct spatial patterns of abnormal neural excitability and neural fragility in AD patients

In healthy elderly brains, neural excitability was highest in the frontal and temporal cortices and was lowest in the central cortices (Fig. 2A), consistent with the anatomical relationships described in epilepsy[33]. In contrast, neural fragility showed a spatial distribution with high fragility over the lateral brain regions in general and reduced fragility over

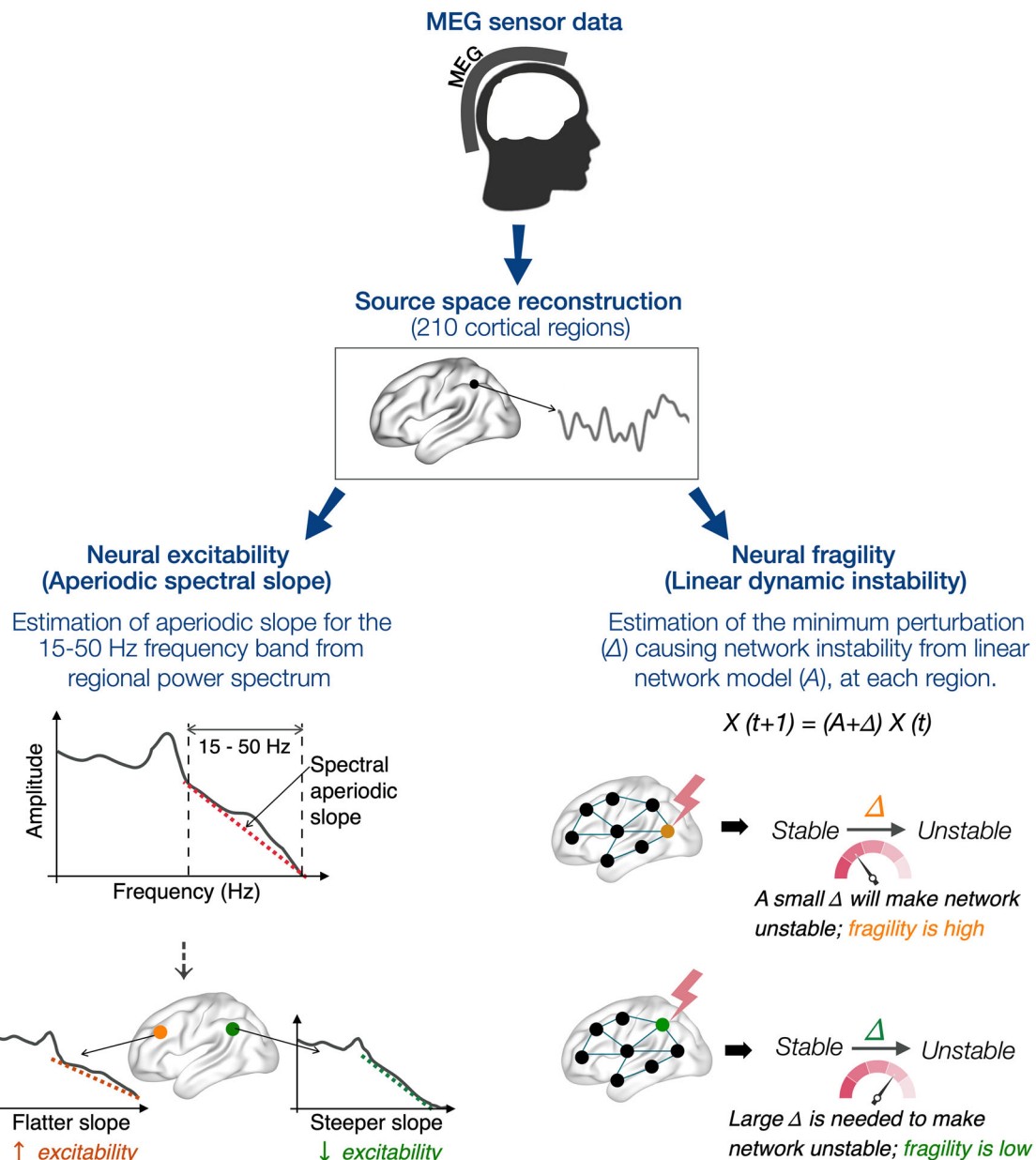

**Fig. 1 | Estimations of neural excitability and neural fragility.** Source reconstructed MEG data at cortical region level (210, Brainnetome-atlas) was used to compute neural excitability (aperiodic spectral slope) and neural fragility (linear dynamic instability). *Neural excitability*: Each regional power spectrum is used to compute the decay aperiodic activity for the frequency band 15–50 Hz. The spectral exponent (slope) indexes the steepness of the decay. A flatter slope, indexed by a higher numerical value (because the slope is negative) represents greater excitability while a steeper slope, indexed by a smaller numerical value represents a lower degree of excitability. *Neural fragility*: The linear network model (*A*) quantifies the network integration over time in each region. Δ is the amount of perturbation added to connections of a given region. The fragility of the node is quantified as the minimal amount of perturbation necessary to push the network from stable to unstable status. A region with small Δ indicates high fragility as a small change can bring instability, while a large Δ indicates low fragility where only a substantial change can induce instability.

the medial regions (Fig. 2A). A hierarchical linear mixed effect model (LMM) with 210 cortical regions nested within each subject, showed no statistically significant associations between neural excitability and neural fragility in healthy elderly, highlighting the dissociable nature of these phenomena under physiological conditions (Fig. 2C; $F = 1.81$, $P = 0.186$). Patients with AD maintained similar spatial patterns of neural excitability and neural fragility distributions albeit both indices showed higher values than controls (Fig. 2D, E). Furthermore, in contrast to healthy elderly, patients with AD showed a positive correlation between neural excitability and neural fragility, indicating that these processes may have shared vulnerabilities in the presence of AD

pathophysiology (Fig. 2F; $F = 5.68$, $P = 0.01$). Importantly, we found that brain regions where neural excitability and neural fragility get altered in AD are spatially distinct. For example, in AD patients compared to controls, neural excitability increased in the posterior temporoparietal and inferior temporal cortices, while neural fragility increased in the medial cortices including the cingulate cortex and dorsal parietal and inferior frontal cortices (Fig. 2G, H). An LMM including age as a covariate showed significantly increased neural excitability and neural fragility in AD compared to controls (Fig. 2I; neural excitability: $t = 3.35$, $P < 0.001$, neural fragility: $F = 4.08$, $P < 0.0001$).

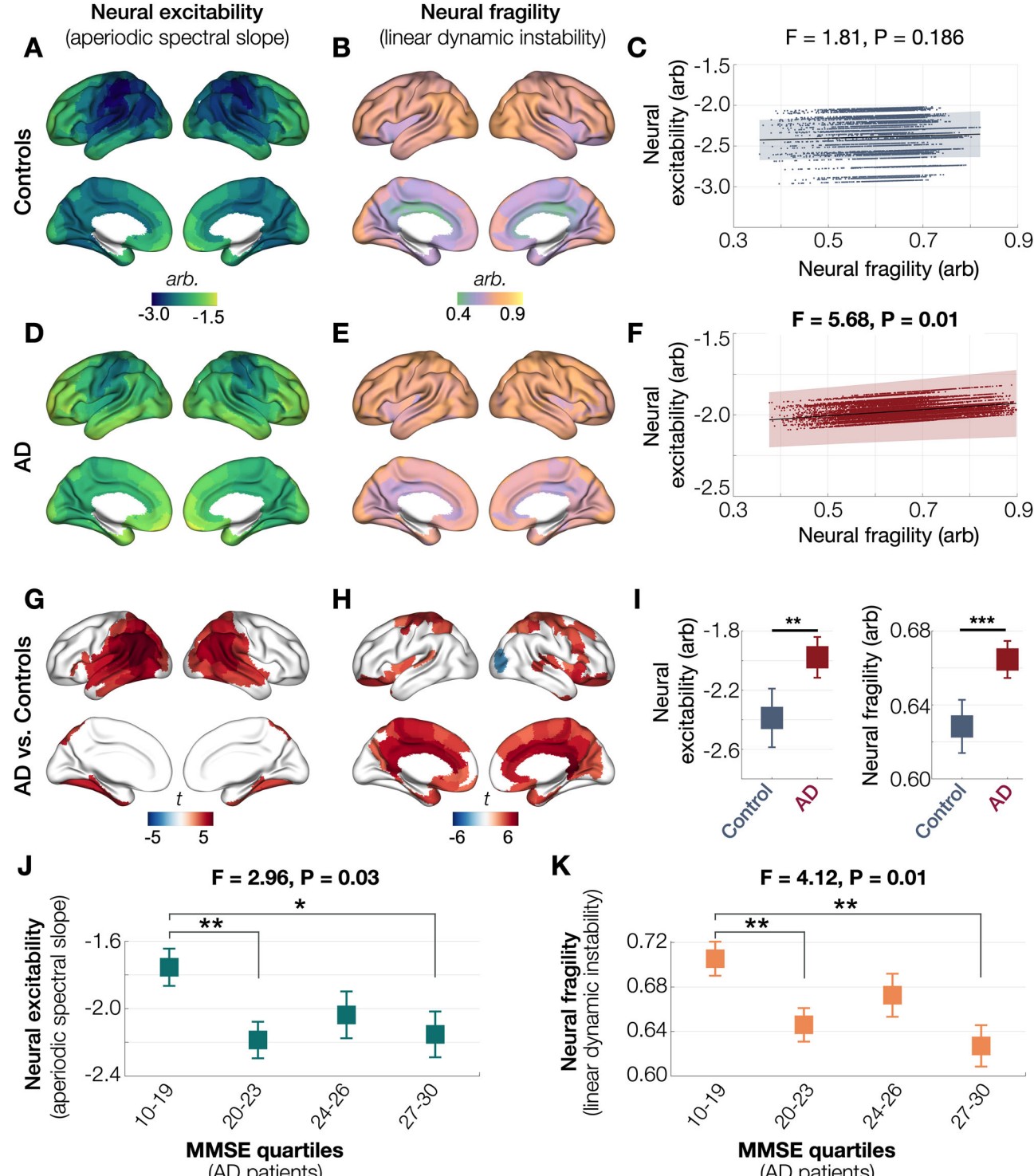

**Fig. 2 | Neural fragility and neural excitability in patients with AD and age-matched controls.** Neural excitability (aperiodic spectral slope) in healthy older adults was highest in the medial and dorsal cortices of frontal and temporal lobes (**A**), while neural fragility (linear dynamic instability) was greater in the dorsolateral than medial cortices (**B**). An LMM analysis showed no associations between neural excitability and neural fragility in the healthy brain (**C**). AD patients retained the same spatial gradient distributions for neural excitability and neural fragility as in controls but showed increased values in both metrics (**D**, **E**). In AD patients, higher neural excitability correlated with higher neural fragility (**F**). The regional patterns of increased neural excitability and neural fragility in AD patients vs. controls involved distinct anatomical regions (**G**, **H**). In a linear mixed model (LMM) with repeated measures, patients with AD showed higher neural excitability (**I**; $t = 3.35$, $P = 0.0011$) and higher neural fragility (**I**; $t = 4.08$, $P < 0.0001$), compared to controls. Global cognitive decline in patients with AD was associated with both increased neural excitability (**J**) and neural fragility (**K**) The $X$ axes of **J** and **K** show the four quartiles of MMSE. The markers depict the least square means corrected for age derived from LMM analyses. In **C** and **F**, the scatter plots indicate each subject's data from 210 cortical regions; the black lines denote the model predictions; the shaded areas indicate the 95% confidence intervals of the model prediction. Brain renderings in **G** and **H** show statistical significance from group comparison between AD and controls after covarying age and thresholded at FDR 5%. The subplots **I**, **J** and **K** denote least square means with the error bars indicating standard error. AD Alzheimer's disease, FDR false discovery rate, LMM linear mixed model, MMSE Mini Mental State Exam (scores range from 0 to 30, with higher scores denoting better cognitive function).

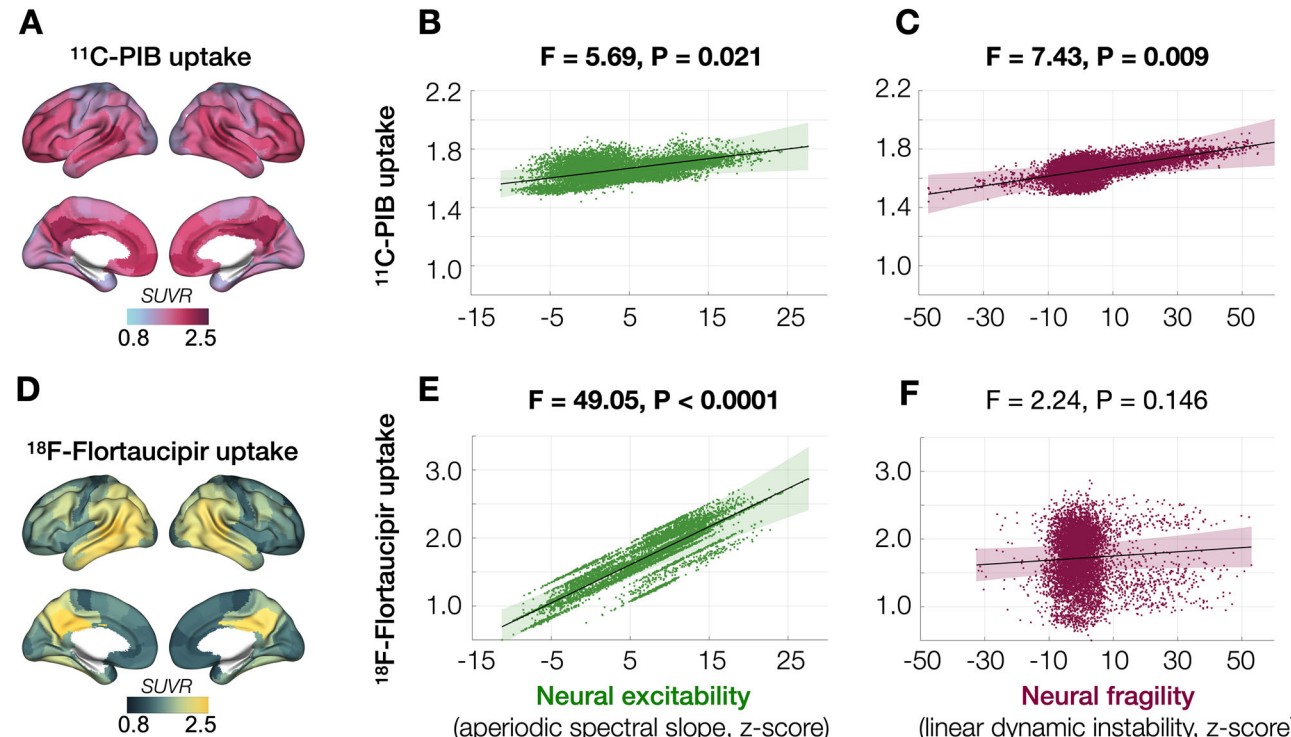

**Fig. 3 | Diverse associations between neural excitability and neural fragility with Aβ and tau accumulations in AD.** Regional Aβ accumulations in patients with AD ($n = 52$) depicted by average uptake of ¹¹C-PIB showed typical regional involvement with higher values in the medial and dorsolateral cortices of frontal and temporal lobes (**A**). A linear mixed model (LMM) analysis revealed that higher Aβ accumulation is positively correlated with both neural excitability (aperiodic spectral slope; **B**) and with neural fragility (linear dynamic instability; **C**). Regional tau accumulation in patients with AD ($n = 35$) depicted by average uptake of flortaucipir showed higher values in temporoparietal cortices (**D**) including inferior temporal, temporoparietal junction, and posterior medial parietal cortices (e.g., retro-splenial, posterior cingulate, and precuneus). In an LMM, higher tau accumulation showed a strong positive correlation with neural excitability (**E**) but was not correlated with neural fragility (**F**). In subplots **B**, **C**, **E** and **F**, the dark line indicates the model prediction, and the shaded area indicates the 95% confidence interval of the model prediction. The scatter plots show each individual subject's data from 210 cortical regions incorporated as repeated measures into the LMM. The models included random intercepts, random slopes, and age and the time difference between MEG and PET as covariates. AD Alzheimer's disease, LMM linear mixed model, PIB Pittsburgh compound B, SUVR standard uptake value ratio.

## Global cognitive deficits in AD are associated with neural excitability and neural fragility

To investigate the relationship between E/I imbalance and cognitive performance, we analyzed how neural excitability and neural fragility relate to Mini-Mental State Exam (MMSE) scores in AD patients. We found that both increased neural excitability and increased neural fragility were significantly associated with lower MMSE scores (Fig. 2J, K; neural excitability: $F = 2.96$, $P = 0.03$; neural fragility: $F = 4.12$, $P = 0.01$). In this analysis, first, for each patient with AD, average neural excitability and average neural fragility were calculated within brain regions that exhibited significant group differences when compared to controls (Fig. 2G, H). Next, using an LMM, these values were modeled against MMSE where patients were categorized into the quartiles of MMSE distribution. Pairwise comparisons revealed that both neural excitability and neural fragility were significantly elevated in the lowest MMSE quartile (Q1: 10–19) compared to the second (Q2: 20–23) and the fourth quartiles (Q4: 27–30). Specifically, neural excitability was higher in Q1 versus Q2 ($t = 2.80$, $P = 0.0065$) and Q1 versus Q4 ($t = 2.18$, $P = 0.0325$); similarly, neural fragility was higher in Q1 versus to Q2 ($t = 2.76$, $P = 0.0072$) and Q1 versus Q4 ($t = 3.17$, $P = 0.0022$). These results suggest that greater E/I imbalance, reflected in both excitability and fragility, is associated with more severe cognitive impairment in AD.

## Distinct associations of neural excitability and neural fragility with tau and Aβ accumulation

To examine the associations between E/I imbalance and AD proteinopathy, we examined the subset of AD patients who were evaluated with multimodal imaging (MEG and PET). Specifically, we quantified the regional tracer uptake of ¹¹C-PIB (Aβ-PET) in 52 individuals and used a hierarchical LMM with random intercepts and random slopes, to examine how Aβ deposition (Fig. 3A) is associated with neural excitability and neural fragility. We found that higher burden of Aβ is positively correlated with both neural excitability and neural fragility, where the latter showed a stronger association (Fig. 3B, C; excitability: $F = 5.69$, $P = 0.021$; fragility: $F = 7.43$, $P = 0.009$). Next, we examined the associations between regional uptake of flortaucipir (Fig. 3D, tau-PET; $n = 35$) and E/I imbalance. LMM analysis revealed that greater tau burden is strongly correlated with higher neural excitability, but not with neural fragility (Fig. 3E, F; excitability: $F = 49.05$, $P < 0.0001$; fragility: $F = 2.24$, $P = 0.146$). The LMMs included subject specific random slopes and intercepts and covariates of age and the time difference between MEG and PET.

As our AD cohort included MCI (CDR = 0.5) and mild AD (CDR = 1) patients, to examine the effect of CDR on these associations, we repeated the LMMs after including CDR as an additional categorical variable. We found that the associations between E/I imbalance and AD proteinopathy remained similar in these models (Supplementary Fig. S1A–D; Aβ-PET: excitability, $F = 5.03$, $P = 0.030$; fragility, $F = 7.18$, $P = 0.011$; tau-PET: excitability, $F = 41.70$, $P < 0.0001$; fragility, $F = 2.38$, $P = 0.136$) while there were no significant interactions of CDR with neural excitability or with neural fragility (Supplementary Fig. S1A–D). Given extensive evidence supporting a primary role of Aβ accumulation—likely preceding tau pathology in the timeline of AD pathogenesis, we also sought to determine that the observed associations

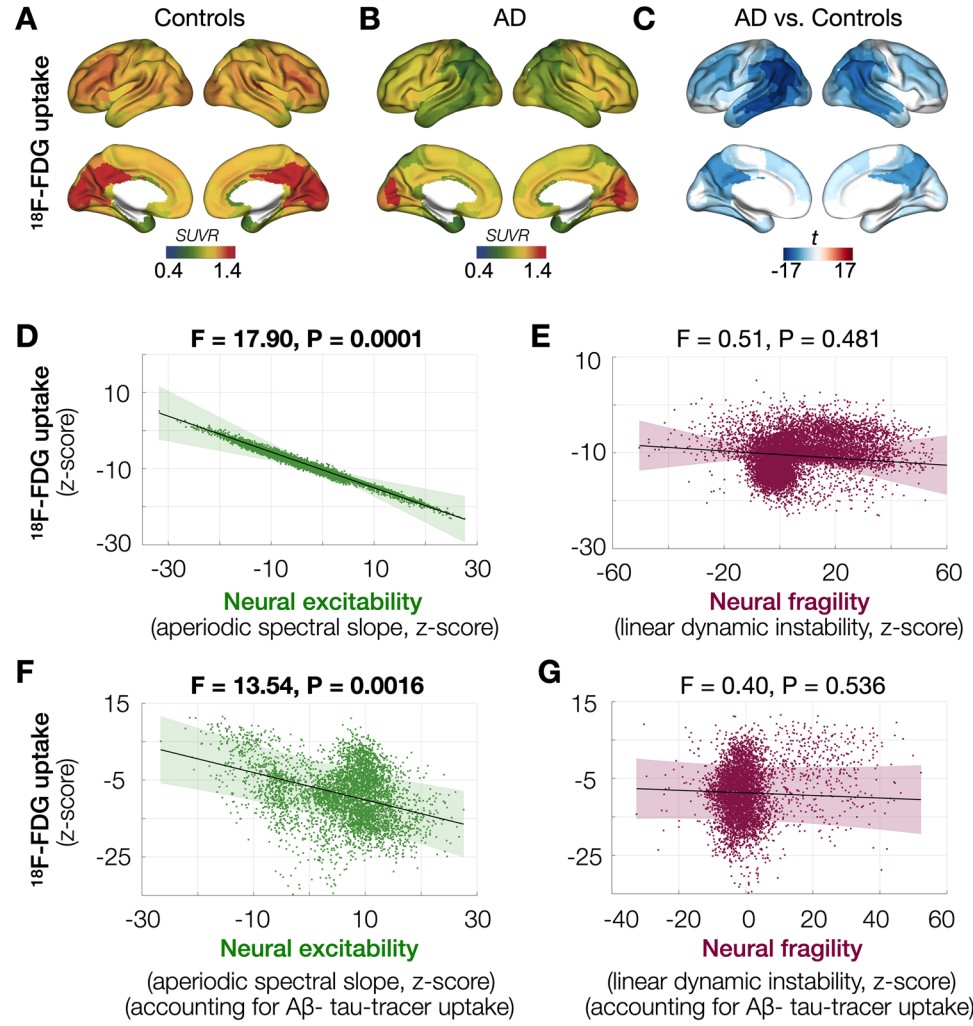

**Fig. 4 | Neural excitability and neural fragility associations with hypometabolism.** Regional glucose metabolism in healthy older adults depicted by average uptake of [18]F-Fluorodeoxyglucose (FDG) showed high uptakes in the medial occipital and posterior cingulate, and moderate uptakes in frontal cortices (**A**). AD patients showed reduced FDG uptake (**B**) with greatest reductions in the posterior temporoparietal cortices compared to controls (**C**). In AD patients imaged with MEG and FDG-PET ($n = 51$), higher neural excitability (aperiodic spectral slope) strongly correlated with greater hypometabolism (**D**), while neural fragility (linear dynamic instability) did not show significant associations (**E**). In AD patients imaged with MEG and triple-PET (Aβ-, tau-, FDG-PET; $n = 23$), higher neural excitability

correlated with greater hypometabolism even after covarying the effects of regional Aβ and tau (**F**), while neural fragility remained uncorrelated (**G**). In subplots D-G, the dark lines indicate model prediction, and shaded areas indicate 95% confidence interval of the model prediction. The scatter plots show each individual subject's data from 210 cortical regions incorporated as repeated measures into an LMM with random intercepts and random slopes. Brain renderings in subplot **C** show statistical significance from unpaired $t$-test between AD and age-matched controls, and thresholded at FDR 5%. AD Alzheimer's disease, FDR false discovery rate, LMM linear mixed model, SUVR standard uptake value ratio.

between E/I imbalance and tau burden were not confounded by Aβ. To test this, we repeated the LMMs for tau while including Aβ burden as an additional covariate, included as centiloids (global Aβ; Supplementary Fig. S2A–B; neural excitability: $F = 47.97$, $P < 0.0001$; neural fragility: $F = 4.07$, $P = 0.054$) and included as regional Aβ (Supplementary Fig. S2C–D; neural excitability: $F = 19.06$, $P = 0.0003$; neural fragility: $F = 0.22$, $P = 0.6462$). These results demonstrated that the association between tau and neural excitability was independent of Aβ.

### Neural excitability but not neural fragility is correlated with hypometabolism

Next, to investigate the associations between E/I imbalance and glucose hypometabolism, which is an index of synaptic dysfunction and potential neuronal loss in AD, we utilized data from a subset of AD patients ($n = 51$) who were imaged with FDG-PET and MEG. Using an age matched normative control group of healthy elderly who were imaged with the same FDG-PET scanning protocol, we first computed

the regional reductions in FDG uptake (z-scores) in AD patients. Consistent with previous reports, posterior temporo-parietal and precuneus regions showed the most severe hypometabolism in AD (Fig. 4A–C). An LMM showed that higher neural excitability is strongly correlated with hypometabolism indicating greater neurodegeneration in AD (Fig. 4D; $F = 17.90$, $P = 0.0001$). In contrast, neural fragility did not show significant associations with brain hypometabolism in AD (Fig. 4E; $F = 0.51$, $P = 0.481$). Next, using a subset of AD patients ($n = 23$), who underwent triple PET imaging (Aβ, tau, and FDG) in addition to MEG, we examined the associations between brain hypometabolism and E/I imbalance after covarying regional Aβ and tau uptakes. We found that, even after controlling for regional Aβ and regional tau, higher neural excitability was positively correlated with brain hypometabolism, indicating that altered local excitability that contributes to E/I imbalance is associated with hypometabolism in AD beyond what was accounted for by regional tau (Fig. 4F; $F = 13.54$, $P = 0.0016$). Neural fragility relationship to FDG-PET uptake remained non-

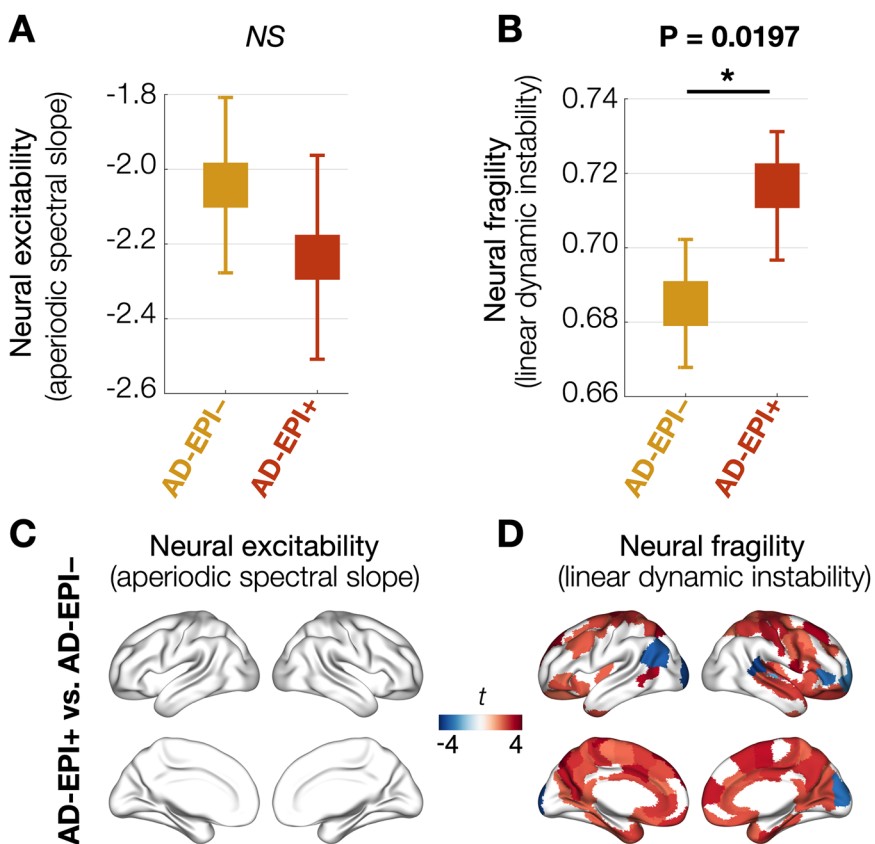

**Fig. 5 | Neural excitability and neural fragility in AD-EPI+ vs. AD-EPI − .** **A** Mixed effects model with repeated measures and AD-EPI+ vs. AD-EPI− group contrast (AD-EPI + , n = 20; AD-EPI − , n = 27) showed that neural excitability (aperiodic spectral slope) was not different between the two groups (**A**; t = 1.08, P = 0.2854). In contrast, neural fragility (linear dynamic instability) was increased in AD-EPI+ compared to AD-EPI− (**B**; t = 2.42, P = 0.0197). The subplots **A** and **B** denote each group's least square means and standard error. The spatial pattern of increased neural fragility in AD-EPI+ patients included bilateral cingulate and medial aspects of frontal and parietal cortices and dorsolateral frontal and temporal regions (**D**). Brain renderings in **C**, **D** show statistical significance from group comparison after covarying age and thresholded at FDR 10%. AD Alzheimer's disease, FDR false discovery rate, AD-EPI + AD patients with epileptic activity, AD-EPI − AD patients without epileptic activity.

significant after covarying Aβ and tau (Fig. 4G; F = 0.40, P = 0.536). The LMMs included random intercepts and slopes, and incorporated age and time difference between MEG and PET as covariates.

We repeated the same statistical models after including CDR as an additional categorical predictor variable and found identical associations between E/I imbalance and hypometabolism where neural excitability, but not fragility, showed significant associations with hypometabolism (Supplementary Fig. S3A–D; excitability, F = 16.96, P = 0.0002; fragility, F = 0.28, P = 0.600), and also after accounting for Aβ and tau in the models (Supplementary Fig. S3C–D; excitability, F = 13.04, P < 0.0020; fragility, F = 0.31, P = 0.582). There were no significant interactions of CDR with neural excitability or with neural fragility when predicting hypometabolism in these models (Supplementary Fig. S3A–D). We also examined for covarying effects of atrophy in these models predicting hypometabolism, by correcting for regional gray matter volumes and found the same statistical relationships where neural excitability showed significant associations with hypometabolism (Supplementary Fig. S4: after correcting for gray matter volume F = 39.43, P < 0.0001; after correcting for tau, Aβ, and gray matter volume: F = 15.09, P = 0.001).

### AD epileptic phenotype showed higher neural fragility but not neural excitability

Next, we compared neural excitability and neural fragility in AD patients who were positive for subclinical epileptic activity (AD-EPI +) as opposed to those who were negative (AD-EPI −). Specifically, this analysis utilized the data from a subset of AD patients (n = 47; Table S3)

who were evaluated with long-term electroencephalography with video monitoring (LTM-EEG) and an hour-long simultaneous MEG and EEG (M/EEG) protocol to identify subclinical epileptiform events. Using a mixed effects model we found that neural fragility is increased in AD-EPI+ than AD-EPI − , while neural excitability did not differ between the EPI subgroups (Fig. 5A, B; excitability: t = 1.08, P = 0.2854; fragility: t = 2.42, P = 0.0197). The models included age and CDR as covariates. The spatial pattern of increased neural fragility in AD-EPI+ involved the medial regions of the brain including much of the cingulate cortex in addition to dorsal parietal and anterior frontal cortices (Fig. 5D). Additional analyses further demonstrated that both AD-EPI− and AD-EPI+ have increased neural excitability as well as neural fragility when compared to controls (Supplementary Fig. S5A–D), which further emphasize the fact that AD epileptic phenotype represents specific abnormalities in E/I imbalance arising from aberrant long-range synaptic input integration processes.

### Discussion

This is a clinical investigation to characterize the associations between aberrant neural circuit activity, AD proteinopathy, and hypometabolism, providing critical insights into the distinct roles of Aβ and tau in E/I imbalance in patients with AD. Our findings demonstrate that E/I imbalance in AD involves aberrant intrinsic neuronal and synaptic processes, as well as deficits in long-range synaptic input integration, indicating abnormalities across different levels of neural circuit organization. In AD patients, Aβ pathology was associated with both types of E/I dysfunction: intrinsic neuronal/synaptic abnormalities,

reflected in altered neural excitability, and deficits in long-range input integration, reflected in increased neural fragility. In contrast, tau pathology distinctly correlated with neural excitability, highlighting its role in aberrant local neuronal and synaptic processes contributing to E/I imbalance. Furthermore, neural excitability was tightly correlated with regional hypometabolism, independent of tau burden. A key finding was the selective increase in neural fragility in AD-EPI+ compared to AD-EPI−, indicating distinct E/I abnormalities associated with the AD epileptic phenotype. These findings highlight the distinct mechanisms of E/I imbalance associated with Aβ and tau pathology and the potential of targeting specific manifestations to enable precise therapeutic interventions to improve cognitive outcomes in AD.

## Diverse manifestations of E/I imbalance in AD

The complex arrangement of brain's functional architecture itself makes E/I imbalance of a given region a multiscale phenomenon, spanning from impaired local neuronal and synaptic activity to impaired integration of long-range inputs. In this study, we used *neural excitability* and *neural fragility* as two complementary measures of E/I imbalance. While both were computed regionally, the two measures differed based on whether they primarily represented local intrinsic neuronal and synaptic processes or long-range synaptic integration processes. Neural excitability, measured as the aperiodic spectral slope (15–50 Hz), has been demonstrated in previous studies as a reliable proxy for excitability of local neuronal populations[25]. A recent study using scalp EEG in combination with in-vivo two-photon calcium imaging in mouse cortex demonstrated that aperiodic spectral slope estimates neural excitability of pyramidal neurons with high accuracy[31]. Previous investigations using human EEG and computational models have also demonstrated that aperiodic, 1/f-like exponent of the power spectrum, in the high frequency range (>15 Hz) in particular, can track shifts in E/I ratio where higher excitatory activity relates to a flattened slope[24,30]. Despite such demonstrated reliability, the development of robust and standardized quantitative measures based on aperiodic slope remains in its early stages, and further methodological refinement and validation of this metric as an index of local hyperexcitability are still needed[25,34]. Neural fragility, the second E/I measure used in this study, estimated the abnormal long-range synaptic input integration[26,27], and was based on the conceptual design that an epileptic event originates within a localized fragile region which renders the cortical circuitry to the brink of instability. A recent clinical study showed high sensitivity of neural fragility to localize the most fragile nodes in patients with epilepsy[26], while an in-vivo epilepsy model further demonstrated its strong associations with time varying aberrant neural activity during epileptogenesis[32]. Specifically, we used the row-perturbation metric of neural fragility, which provides a perturbation-based framework to quantify linear dynamic instability as an estimate of dysfunction in long-range synaptic input integration by a local neuronal population[27]. Importantly, we show that in cognitively healthy older adults, neural excitability and neural fragility were not correlated with each other, suggesting that these phenomena are relatively independent under physiological conditions. In contrast, neural excitability and neural fragility were positively correlated in patients with AD, indicating that AD pathophysiological mechanisms may influence both E/I mechanisms—local neuronal hyperexcitability and impaired long-range input integration. A key finding from the current study is that E/I imbalance in AD involves impairments at multiple levels along the functional architecture of neural circuits.

## Epileptic manifestations signify an AD phenotype with greater neural fragility

The prevalence of subclinical epileptic events and silent seizures reported in clinical AD cohorts ranges from 22–54% (depending on the sensitivity of assays), as opposed to less than 11% reported in healthy elderly populations[12,14,35,36]. The signature increase of neural fragility in

AD-EPI+ compared to AD-EPI− suggests that the epileptic phenotype in AD represents a subset of patients who harbor greater impairments in long-range synaptic input integration processes. It is important to note that neural excitability is similarly elevated in both AD-EPI+ and AD-EPI− patients, exhibiting a generalized E/I imbalance in AD primarily contributed by local neural and synaptic dysfunctions. Collectively, these findings provide a neurophysiological basis for a seemingly contradictory observation from clinical investigations: while studies using quantitative assays have suggested a generalized increase in E/I in AD as indicated by flattened aperiodic slopes[37,38], several carefully constructed observational studies have consistently found epileptic events in only a fraction of AD patients[12,14,35,36]. The current results present an interpretation to these incongruent outcomes by demonstrating the heterogeneous manifestations of E/I imbalance in AD, where local hyperexcitability appears to be a more generalized feature, while specific deficits in long-range synaptic integration may predispose the epileptic phenotype. Previous studies showing reliable clinical distinctions between AD-EPI+ vs. AD-EPI− including earlier age of onset and more aggressive disease course[12,15,39], further corroborates the phenomenon of AD-EPI+ as a specific phenotype with abnormalities in distinct E/I processes. A relevant clinical implication of the AD-epileptic phenotype was further highlighted in a recent clinical trial showing that only AD-EPI+ patients benefited from antiepileptic treatment with levetiracetam as shown by improved secondary endpoints defined by cognitive control ability and visuospatial memory[40].

Both animal studies and clinical investigations have tried to tease apart the associations between epileptic manifestations and AD proteinopathy. Seizures or interictal epileptic activity in AD transgenic mouse models have been identified with both Aβ and tau, independently. For example, mice that overexpress hAPP exhibit reduced latency to seizure onset, increased incidence of seizures, and higher seizure-associated deaths[4,5,20,41], which are often found in the absence of tau pathology. While transgenic models that primarily exhibit tau pathology with limited or no Aβ deposition show less consistent epileptic manifestations, a clear role of tau in E/I imbalance has been demonstrated in studies where genetic reduction of tau expression is associated with reduced hyperexcitability and stabilization of neural networks[6,42,43]. Conversely, in clinical studies, epileptiform activity in AD patients is most frequently found either preceding or coinciding with the clinical symptoms[14,16,44], hence closely aligning with the time frame where dual proteinopathy is evident in AD. A recent clinical study showed that AD-EPI+ patients, compared to AD-EPI− patients, have higher bilateral Aβ burden particularly in the medial temporal, lateral temporal, frontal, and medial parietal regions, as well as asymmetrical tau burden, which is higher in the epileptogenic hemisphere[45]. A case series in AD patients, using foramen ovale electrodes, also demonstrated greater asymmetrical tau deposition in the epileptic temporal lobe[46]. Together, these results indicate that both pathological Aβ and tau likely contribute to the development of AD epileptic phenotype. While we did not directly examine the role of Aβ and tau in AD-EPI+ phenotype, our study offers insights into how AD dual proteinopathy may disrupt neural circuit excitability. First, we demonstrate that epileptic phenotype in AD is characterized by increased neural excitability compared to the healthy aging brain, and by greatly elevated neural fragility, surpassing what is observed in AD-EPI− patients. Next, we show that neural excitability correlates with both Aβ and tau pathology, while neural fragility selectively correlates with Aβ burden. These findings support the notion that both Aβ and tau may contribute to epileptiform activity in AD, albeit through distinct mechanisms of E/I imbalance.

## Distinct associations of Aβ and tau with E/I imbalance in AD

Our findings demonstrate distinct associations of Aβ and tau with diverse markers of E/I imbalance, offering complementary clinical evidence to mechanistic models of AD. We found that Aβ pathology

was associated with both increased neural excitability and elevated neural fragility, whereas tau selectively correlated with increased neural excitability but not with fragility—suggesting divergence in the pathological roles of Aβ and tau in E/I imbalance. The finding that hypometabolism is also selectively associated with neural excitability is also consistent with the widely known tight association between tau and hypometabolism from PET studies[47]. These findings are consistent with mechanistic insights from in-vivo and in-vitro AD models. Although the mechanisms by which Aβ and tau exert E/I imbalance remains an area of active research, several recent advances have clearly demonstrated that these effects are distinct and diverse. For example, AD models have shown that pathological Aβ inhibits extracellular glutamate reuptake[48] and interferes with GABAergic inhibitory neurotransmission including downregulation of Nav1.1 channels in parvalbumin-positive interneurons[4,49]. Unlike Aβ, tau is involved in a variety of complex physiological functions, and its pathological impact is thought to arise both from abnormal gain-of-function mechanisms and by facilitating other disease pathways[50]. The cellular effects of tau are also highly diverse, including direct disruption of pre- and post-synaptic transmission and activation of aberrant immune responses[6,23,51]. These mechanisms, either acting independently or together, ultimately may impair intrinsic firing properties of neurons. Despite demonstration of these multiple potential pathomechanisms, the significance of such distinctions in truly disease-relevant contexts cannot be shown in animal models. Our findings not only help bridge this crucial gap in knowledge but also raise important questions that warrant further investigation. For example, the findings that neural excitability, which indexes local neuronal and synaptic hyperexcitability being closely aligned to tau accumulation poses the intriguing question whether altered neural excitability is a trigger for the spread of tauopathy or whether it is a consequence of pathological tau spread. Animal models have indeed shown that aberrant intrinsic excitability can contribute to pathological tau accumulation[52–54], although this has yet to be investigated in human patients. Another open question is whether Aβ accumulation—which occurs simultaneously across multiple brain regions in AD[55]—is associated with the cytoarchitectural properties of long-range neuronal connections. A deeper understanding of how Aβ and tau affect E/I imbalance in clinical AD populations will enable targeting their specific mechanisms in novel treatments.

The differential relationships of Aβ and tau to E/I imbalance in AD have important clinical implications for protein lowering therapies. Currently available anti-Aβ immunotherapies offer about a30% benefit in slowing cognitive decline compared to placebo, yet with substantial heterogeneity in clinical response. It remains unclear whether Aβ- or tau-associated E/I imbalance influence these therapeutic outcomes. It is possible that patients may exhibit a poor clinical response, because the specific pathogenic pathways that contribute to E/I imbalance either remain unaffected by the treatment or interact negatively with it. Determining whether antiepileptic drugs could serve as effective adjunct therapies to improve clinical outcomes in these patients is of high clinical significance. Furthermore, the precise cellular mechanisms targeted and normalized by current immunotherapies are not fully understood. While these therapies primarily engage Aβ-related molecules, baseline tau burden has also been identified as a key predictor of clinical outcomes. Neural fragility and neural excitability, as shown in the current study, are useful biological indices to quantify the functional consequences of Aβ and tau in AD patients and could help determine which pathological factor—Aβ or tau—more effectively contributes to clinical improvements when treated with anti-Aβ immunotherapies.

## E/I imbalance as a core element of neural circuit dysfunction

Functional impairment of brain activity in AD has been demonstrated at multiple scales within the hierarchical organization of the nervous system from individual neurons to large-scale canonical networks. Recent multimodal imaging studies—integrating molecular imaging using Aβ- and tau-PET with functional modalities such as MEG, EEG, and fMRI—have demonstrated distinct network-level abnormalities associated with AD proteinopathy. For example, increased low-frequency oscillatory activity is strongly correlated with Aβ pathology, whereas reductions in alpha-band oscillations are more closely associated with tau deposition[56–59]. Moreover, impaired oscillatory synchrony patterns, as well as functional connectivity abnormalities estimated from fMRI, have been identified as important determinants of tau spread in individuals along the AD neuropathological spectrum[60,61]. While these network-level manifestations complement the cellular level mechanisms from animal studies, a coherent translational framework that integrates these multiscale findings has remained elusive. A recent review highlighted how E/I imbalance may serve as a unifying neurophysiological framework that integrates findings across scales of functional impairment[62]. Furthermore, E/I imbalance constitutes a core element in AD pathophysiology, not only encompassing the impact of proteinopathy but also involving other critical processes that appear to share a complex relationship with hyperexcitability, such as vascular dysfunction[63], neuroinflammation[53], and cholinergic impairment[64]. The significant association between neural excitability and hypometabolism, even after accounting for regional AD proteinopathy in our results, supports the concept of potential relationships between E/I imbalance and other contributory pathomechanisms in AD. In broader terms, E/I imbalance is a fundamental response to disrupted neural circuits in any neurological condition and may represent maladaptive as well as compensatory processes. Quantitative measures to accurately characterize specific E/I abnormalities such as neural excitability and fragility, and their associations with the pathological markers of Aβ and tau, therefore may play an essential role in identifying therapeutic targets and in evaluating the efficacy of interventions.

## Limitations

The current study is not without limitations. The current AD cohort is predominantly early-onset, and future studies replicated in larger, more diverse cohorts, will help generalize the findings to the more common late-onset AD population and to examine associations with genetic and lifestyle factors. Although we show robust associations of neural fragility and neural excitability with protein aggregates in our multimodal imaging cohorts, we did not have Aβ-PET and tau-PET imaging in all our sub-cohorts, which were characterized for epileptiform manifestations as AD-EPI+ and AD-EPI − . Future studies including AD patients evaluated with detailed seizure semiology using M/EEG, plus Aβ- and tau-PET will help determine the specific regional associations of E/I imbalance and AD proteinopathy in AD epileptic phenotype. It is also noteworthy that while the spectral aperiodic slope has shown promise as a reliable proxy for local neuronal and synaptic activity, the development of quantitative frameworks for its use is still evolving. Our findings underscore the urgency of establishing standardized methodologies for aperiodic slope analyses to enable more consistent interpretations and accelerate progress in the field. Non-invasive electrophysiology has limited ability to provide cellular-level interpretations of neural excitability and neural fragility, and future studies utilizing biophysical models and parallel animal models could address this gap by parsing the excitatory and inhibitory neuronal contributions to these metrics.

In summary, using multimodal imaging of high-resolution electrophysiology and PET in patients with early-stage AD, we demonstrated that Aβ and tau are likely associated with altered E/I balance through different mechanisms. While tau is distinctly associated with the indices of altered E/I that primarily represent local neuronal and synaptic excitability dysfunction, Aβ is correlated with both the indices of impaired long-range synaptic input integration and those of local

excitability dysfunction. Importantly, the AD-EPI+ phenotype represented greater impairments in long-range synaptic input integration processes. The findings elucidate diverse mechanisms underlying E/I imbalance in AD and underscore the critical importance of targeting the distinct pathogenic effects of Aβ and tau to develop precise and effective therapies.

## Methods
### Participants
A total of 82 patients with AD and 40 age-matched controls were included in this study (Table 1). All patients fulfilled the current diagnostic criteria for probable AD or MCI due to AD[65,66] and were positive for AD biomarkers, and represented early stage disease (Clinical Dementia Rating scale, CDR 0.5 or 1). Patients with AD and controls did not differ on sex, race or education (Table 1). All participants were recruited from research cohorts at the University of California San Francisco (UCSF) Memory and Aging Center. Each participant underwent a complete clinical history, physical examination, neuropsychological evaluation and brain magnetic resonance imaging (MRI). In addition, each participant also underwent a minimum 10-minute session of magnetoencephalogr (MEG) recording at rest. Clinical diagnosis for patients with AD was established by consensus in a multidisciplinary team. Subsets of patients underwent positron emission tomography (PET) with: (1) Aβ-specific radiotracer [11]C-PIB (n = 52); (2) tau-specific radiotracer [18]F-flortaucipir (35); (3) [18]F-labeled fluorodeoxyglucose (FDG, n = 51); (4) all three tracers (n = 23). A subset of AD patients (n = 47) were evaluated with two forms of sensitive neurophysiological monitoring: extended/long-term electroencephalography (LTM-EEG) and MEG with simultaneous EEG (MEG-EEG), to determine the presence of subclinical epileptiform activity. Clinical and demographic characteristics of this sub-cohort are presented in Table S3. Twenty AD patients (40%) were identified as positive for subclinical epileptiform activity (AD-EPI+) and 27 patients were identified as AD-EPI−. All patients were carefully screened for the absence of any overt epileptic events or seizures as we were specifically targeting subclinical level epileptic manifestations in this cohort. None of the patients were on antiepileptic treatment. Lobar level distribution of epileptic activity in AD-EPI+ patients has been presented in a previous study[15]. The age-matched controls were recruited from the community and the eligibility criteria included normal cognitive performance, normal MRI, and absence of neurological, psychiatric, or other major medical illnesses. In addition, we also utilized an amyloid negative normative dataset from the Lawrence Berkeley National Laboratory (LBNL) to estimate the normalized scores of FDG uptake data (n = 53 with average age = 61.36 ± 7.76). Informed consent was obtained from all participants or their assigned surrogate decision makers. The study procedures were approved by the Institutional Review Boards (IRB) at UCSF, LBNL, and University of California Berkeley, and were carried out in accordance with the IRB for human studies and the guidelines proposed in the Declaration of Helsinki.

### Neuropsychological assessment
For each participant, a structured caregiver interview was used to assess the CDR[67]. Each participant was also assessed via a Mini Mental State Examination (MMSE) and a standard battery of neuropsychological tests[68].

### Resting state MEG data acquisition and analysis
Each subject underwent MEG recording on a whole-head biomagnetometer system consisting of 275 axial gradiometers (MISL, Coquitlam, British Columbia, Canada). Three fiducial coils including nasion, left and right pre-auricular points were placed to localize the position of the head relative to the sensor array, and were later co-registered to each individual's respective MRI to generate an individualized head shape. Data collection was optimized to minimize within-session head movements, keeping them below 0.5 cm. A minimum of 10 minutes continuous recording was collected from each subject while lying supine and awake with eyes closed (sampling rate: 600 Hz). We selected a 60-s (1 min) continuous segment with minimal artifacts (i.e., minimal excessive scatter at signal amplitude <10 pT), for each subject for analysis. The study protocol required the participant to be interactive with the investigator at the beginning of the data collection, affirming awake status. Spectral analysis of each recording was visually inspected and those showing known electrophysiological features of sleep were excluded. Artifact detection was confirmed by visual inspection of sensor data, and channels with excessive noise within individual subjects were removed prior to analysis. Resting state MEG data were pre-processed using the Fieldtrip toolbox, and source space reconstruction was performed using custom-built MATLAB software tools[57]. We estimated the neural metrics of E/I imbalance (see below) for 210 cortical regions represented in the Brainnetome atlas and excluded the subcortical regions. This approach enabled the use of source space reconstructed data with high accuracy.

**Neural excitability.** To quantify neural excitability in each region, we used the spectral aperiodic slope within 15−50 Hz band. In each region, the aperiodic spectral component, $PSD_{aper}(f) \sim 1/f^{\alpha}$, was extracted using a three-step least-square procedure[29], and the spectral exponent $\alpha$ was evaluated as the slope of the linear regression of spectral-power in log(spectral-power)-log(f) plot. Previous studies have suggested that the aperiodic spectral slope of higher frequency range is reflective of local neuronal excitability, where a flatter aperiodic slope, indicated by a higher numerical value (since the slope is negative), especially in the frequency range (>15 Hz), has been shown to indicate greater neural circuit excitability[30]. Based on these findings, we used the frequency range of 15−50 Hz to compute aperiodic slope.

**Neural fragility.** To quantify neural fragility, we extracted the linear dynamics of cortical regional networks for each 100 ms time window (50% overlap) from the 210 source-localized regional time courses. We then evaluated the magnitude of perturbations required to destabilize these linear dynamics. The linear dynamics were modeled by $x(t+1) = A_j x(t)$ where $x(t) \in R^{210}$ represents the regional time courses, and $A_j \in R^{210 \times 210}$ is the time evolution matrix, where $j$ denotes the index of time windows. The matrix $A_j$ was estimated using linear regression on the data within each time window. A linear system becomes unstable when a perturbation causes a change in the time evolution matrix. This change can be modeled as an additive perturbation $\Delta$ to the matrix: $A_j \rightarrow A_j + \Delta$. Specifically, we considered row perturbations[27] as the structure of the perturbation. Depending on which rows (regions) are preferentially affected, different perturbation strengths are required to render the network unstable. In this framework, neural fragility is defined as the magnitude of the minimum perturbation needed to make the network unstable. For each estimated $A_j$, we computed the minimum norm perturbation $\Gamma_{jk}$ for the $k$-th column perturbation. The corresponding fragility metric at each region $k$ was then evaluated as $||\Delta_j||_k = 1 - \Gamma_{jk}/\max(\Gamma_j)$, where a value of 1 indicates the most fragile. The fragility metric $||\Delta_j||_k$ can be visualized as a time-vs-region heatmap. To quantify regional neural fragility, we computed the average of the fragility metric over time, $\langle ||\Delta||_k \rangle$.

### Detection of subclinical epileptiform activity
A subset of 47 patients underwent overnight LTM-EEG telemetry, followed by one-hour M/EEG (21-lead EEG) the following day. LTM-EEG telemetry: Patients were evaluated at the Clinical Translational Science Institute Research Center at Moffitt Hospital at UCSF. Monitoring included overnight long-term recording using silver cup electrodes in the standard international 10−20 electrode array, with 3 minutes of hyperventilation. M/EEG: One-hour resting-state M/EEG was

performed at the UCSF Biomagnetic Imaging Laboratory. The MEG was recorded using the same system described for 10 min scans above. LTM-EEG and M/EEG and were read by experienced epileptologists (P.A.G and H.E.K) and clinical neurophysiologists (K.G.R and K.V). The Spike Density Calculation Engine in Persyst-11 EEG software was used to help detect epileptiform activity on the LTM-EEG. Epileptiform activity on LTM-EEG or M/EEG was defined using the same criteria mentioned in our previous study[12] and included the presence of spike or sharp wave that fulfilled the specific criteria to be identified as an interictal epileptiform discharge (i.e., abrupt change in polarity with specific morphological spike and sharp wave characteristics that is clearly distinguished from background activity; a clear physiologic field; disrupted background associated with subsequent slowing or embedded in bursts of focal slowing)[69,70]. Normal EEG variants such as small sharp spikes and wicket spikes were not included as subclinical epileptiform activity. Entire recordings were reviewed by visual inspection, and determination of epileptiform activity was made by consensus between the clinicians who reviewed the recordings (P.A.G, H.E.K, K.G.R, and K.V for LTM-EEG; H.E.K and K.G.R for M/EEG).

## PET Data Acquisition and Image Processing

Detailed descriptions of flortaucipir, PIB-PET, and FDG-PET acquisition are available in previous publications[71,72]. All PIB-PET scans were acquired at LBNL on a Siemens Biograph 6 Truepoint PET/CT scanner (Siemens Medical Systems) in 3D acquisition mode. Attenuation correction was performed using a low-dose CT/transmission scan acquired prior to the scans. For PIB-PET, dynamic acquisition was performed for 90 min (35 frames total) immediately after intravenous injection of ~15 mCi of 11C-PiB, data from 50–70 min were normalized to a cerebellar gray matter reference region to calculate SUVRs. All FDG-PET, scans were acquired at LBNL, including 39 scans on a Siemens Biograph 6 Truepoint PET/CT scanner (Siemens Medical Systems; in 3D acquisition mode) and 12 scans on ECAT EXACT HR (3D acquisition mode). A 30-min scan ($6 \times 5$ min frames) was acquired 30 min following i.v. injection of 5–10 mCi of 18F-FDG. All flortaucipir PET scans were acquired on a Siemens Biograph PET/CT scanner (Siemens Medical Systems) in 3D acquisition mode. 31 flortaucipir PET scans were acquired at LBNL and 4 scans were acquired at the UCSF China Basin Imaging center. Flortaucipir Participants were injected with 10 mCi of tracer and scanned 80–100 min post-injection ($4 \times 5$ min frames). PET images were reconstructed using an ordered subset expectation maximization algorithm with weighted attenuation and smoothed with a 4 mm Gaussian kernel with scatter correction. Averaged, smoothed PET scans were co-registered to their respective structural MRIs and intensity normalized according to average tracer-specific reference regions to generate standardized uptake value ratio (SUVR). The tracer specific reference regions included, cerebellar gray matter for PIB-PET ($SUVR_{50-70}$), inferior cerebellar gray matter for tau-PET ($SUVR_{80-100}$) and pons for FDG-PET ($SUVR_{30-60}$). PET SUVR images were warped to the Montreal Neurological Institute space using the deformation parameters generated by the SPM12 segmentation of the respective structural MRI. We also calculated a neocortical composite SUVR from Aβ-PET and converted to centiloid values using previously described methods[73].

## Magnetic resonance image acquisition and analysis

Structural brain images were acquired from all participants using a unified MRI protocol on a 3 Tesla Siemens MRI scanner at the Neuroscience Imaging Center at UCSF. Structural MRIs were used to generate individualized head models for source space reconstruction of MEG sensor data. The structural MRI scans were also used in the clinical evaluations of patients with AD to identify patterns of gray matter volume loss to support the diagnosis of AD. For the 51 subjects who were imaged with MEG and FDG-PET, we also computed the gray matter volumes for each of the 210 cortical regions defined in the

Brainnetome atlas using SPM-12. The regional gray matter volumes were calculated using the morphometry pipeline implemented in the CAT12 toolbox with default parameters. All MRI images used in the volumetric analysis ($n = 51$) were acquired within 12 months of the MEG scan (average time difference in months, $0.18 \pm 0.40$).

## Statistical models

**Hierarchical linear mixed effects models.** To examine the relationships between neural excitability and neural fragility, we used hierarchical linear mixed effects models (LMMs), in AD patients and in controls. Specifically, fragility and excitability values were entered into the model per each cortical region (210) per subject. We used the MIXED procedure in SAS (PROC MIXED) and included random intercepts and random slopes. As previous studies have shown that aperiodic slope (neural excitability) is correlated with age, we included age as a covariate in the models. The models also included subject identity and included 210 cortical regions as repeated measures. A separate model was used for each group.

To examine the associations between E/I imbalance and protein tracer uptake from PET imaging, we used similar hierarchical mixed models (PROC MIXED) as described above using regional neural fragility and neural excitability. All models included a repeated-measure structure to incorporate data from 210 cortical regions per subject. Specifically, the models included both random intercepts and random slopes to model subject-specific variation. In repeated measures data, observations within the same individual are inherently correlated; modeling random effects allows LMMs to appropriately account for this intra-subject correlation, thereby avoiding inflated Type I error rates that would arise from treating these observations as independent. All models included age and the time difference between MEG and PET imaging as covariates. The distribution of residuals from LMMs was examined using histograms and evaluated for skewness to confirm that the model assumptions were satisfied. First, to eliminate the confounding effect of the physiological regional variability of neural excitability and neural fragility in the correlation analyses, we computed z-maps of each metric (neural fragility and neural excitability) for each AD patient based on the age-matched control data. The LMM model examining the associations with Aβ accumulation included regional values of neural excitability and neural fragility as predictor variables and regional SUVR of [11]C-PIB (Aβ tracer uptake) as the dependent variable. The LMM model examining the associations with tau accumulation included regional values of neural excitability and neural fragility as predictor variables and regional SUVR of flortaucipir as the dependent variable. To examine the relationships with neuronal hypometabolism (FDG-uptake patterns), we first computed the z-scores of regional [18]F-FDG SUVR for each AD patient. This was an essential step to eliminate the confounding effect of the physiological regional variability of metabolism. The z-scores of regional FDG-uptake values were generated for each subject based on an age-matched normative database ($n = 53$ with average age $= 61.36 \pm 7.76$). To examine the relationship between E/I imbalance and FDG-uptake after controlling for tau and Aβ, we included regional tau and Aβ tracer uptake values as additional covariates in the models.

We repeated all models, including CDR as an additional categorical predictor variable, to examine the associations of CDR = 0.5 vs. CDR = 1. As both clinical and animal model studies provide evidence to support the phenomenon that Aβ likely has an initial role in the development of AD proteinopathy, we repeated the LMMs which examined the associations of tau accumulation, after controlling for Aβ. Specifically, in the full cohort of AD patients imaged with tau-PET ($n = 35$), we included the global Aβ burden, estimated as centiloids, as a covariate in the model. Additionally, in the subset of AD patients who were imaged with tau-PET and [11]C-PIB (as a uniform Aβ-PET tracer; $n = 23$), we included regional [11]C-PIB SUVR as a covariate. Because hypometabolism has been shown to correlate with gray matter

atrophy in AD patients, we repeated the LMMs which examined the associations with FDG-uptake after controlling for cortical atrophy by including regional cortical gray matter volume (210 regions) as an additional covariate.

To examine the group differences between cohorts (controls, AD, AD-EPI +, AD-EPI − ), we used hierarchical LMMs including regional values of neural fragility and neural excitability to examine group effects. For example, neural fragility was the dependent variable, with group identity (e.g., AD or control) as a categorical predictor and age as a covariate. The model included 210 cortical regions as repeated measures, with random intercepts and random slopes. An identical model was used for neural excitability. Similar models were used to examine the group differences of neural excitability and neural fragility in the contrasts of AD-EPI+ vs. controls, and AD-EPI− vs. controls. To compare between AD-EPI− vs. AD-EPI +, we used identical models using z-scores of neural fragility and neural excitability including CDR as an additional covariate. We report the least-square means, confidence intervals and statistical significance from these analyses.

To examine the associations between E/I imbalance and global cognitive decline, we used LMMs based on the quartiles of MMSE distribution in AD patients. Specifically, we categorized each AD patient into one of four MMSE stages depicting their global cognitive dysfunction. The four stages of MMSE were identified by the quartile boundaries and included 30–27, 26–24, 23–20, and 19–10. Two separate mixed models included neural excitability and neural fragility as dependent variables and MMSE quartile as a categorical predictor variable. For each individual subject, we computed the average neural excitability and neural fragility within the respective regions that were identified as significantly elevated in the group contrast between AD and controls (Fig. 2G, H). Because neural excitability and neural fragility were correlated in AD patients, the LMM for neural excitability included neural fragility as a covariate, whereas the LMM for neural fragility included neural excitability as a covariate. Both models also included age and time difference between MEG and MMSE evaluation as additional covariates. With observations stratified into meaningful epochs of disease severity this analytic approach provided precise estimates of error within these categories and allowed robust detection of change between them.

### Spatial maps of group analyses

To examine the group differences in neural excitability and neural fragility in AD vs. controls, AD-EPI+ vs. controls, AD-EPI− vs. controls and AD-EPI+ vs. AD-EPI − , we used analysis of covariance models including age as a covariate. We report the t-values of the estimated group effects, adjusted for age. All brain renderings depicting group averages and contrasts were generated using BrainNet Viewer[74].

### Reporting summary

Further information on research design is available in the Nature Portfolio Reporting Summary linked to this article.

## Data availability

All data associated with this study are presented in the main content and supplemental information sections. Source data for figures are provided with this paper. Deidentified imaging (MEG and MRI) data are saved in the OSF and are publicly available. (https://osf.io/pd4h9/files/osfstorage). Deidentified PET and clinical data will be shared on request from qualified investigators for the purposes of replicating procedures and results, and for other non-commercial research purposes within the limits of participants' consent. Correspondence and material requests should be addressed to Kamalini.ranasinghe@ucsf.edu. Source data are provided with this paper.

## Code availability

All custom scripts used for data analysis in this study have been deposited on a GitHub repository (https://github.com/kamalinigr/megabtaufdg).

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

## Acknowledgements

The authors thank the patients and their caregivers for their participation in this research. Avid enabled the study by providing tracer flortaucipir but did not contribute to study design or data analysis/interpretation. This study was supported by the National Institutes of Health grants: R01AG088398–01 (K.G.R.), K08AG058749 (K.G.R.), R21AG077498-01 (K.G.R.), R21DC021557 (S.S.N.), K23AG038357 (K.V.), R01AG045611 (G.D.R.); R01AG062422 (G.D.R.); R01AG034570 (W.J.), R01AG062542 (W.J.); RF1NS100440 (S.S.N.), R01DC017091 (S.S.N.), R01AG062196 (S.S.N.), P50DC019900 (S.S.N.), R01DC021711 (S.S.N.), R24MH120037 (S.S.N.); a grant from John Douglas French Alzheimer's Foundation (K.V.); grants from Larry L. Hillblom Foundation: 2015-A-034-FEL and (K.G.R.); 2019-A-013-SUP (K.G.R.); a grant from Alzheimer's Association: AARG-21-849773 (K.G.R.).

## Author contributions

K.G.R., K.V., G.D.R. and S.S.N. conceptualized and designed the study, interpreted the results and contributed to manuscript writing. B.L.M., K.V., K.P.R., G.D.R., P.A.G., H.E.K. and J.H.K. contributed to design, analysis, and interpretation of clinical data. K.G.R., K.K., K.P.R., P.A.G., H.E.K., C.Y., F.S., K.V., G.D.R., J.H.K. and W.J. contributed to data analysis and interpretation of results. F.S. and C.Y. contributed to data collection, preprocessing, and data analyses.

## Competing interests

K.G.R., K.K., K.P.R., P.A.G., H.E.K., C.Y., F.S., K.V., G.D.R., J.H.K. and W.J. declare no competing interests relevant to this work. S.S.N. serves as a founding board member in Hippoclinic Inc., as a private consultant to MEGIN Inc. and is the PI for an industry contract from Ricoh MEG USA. B.L.M. has the following disclosures: serves as Medical Director for the John Douglas French Foundation; Scientific Director for the Tau Consortium; Director/Medical Advisory Board of the Larry L. Hillblom Foundation; and Scientific Advisory Board Member for the National Institute for Health Research Cambridge Biomedical Research Center and its subunit, the Biomedical Research Unit in Dementia, UK.
