## [Transparent Peer Review file · Nature Communications]

Distinct manifestations of excitatory-inhibitory imbalance associated with amyloid- β and tau in patients with Alzheimer's disease

Corresponding Author: Dr Kamalini Ranasinghe

Version 0:

Reviewer comments:

Reviewer #1

(Remarks to the Author)
Please, see attached document.

Reviewer #2

(Remarks to the Author)
Summary

The authors aim to quantify excitatory-inhibitory (E/I) imbalance in AD using MEG and subsequently correlate these measures with AD pathology and epileptic activity. They examine two E/I, termed "neural-excitability" (aperiodic slope of the power spectrum) and "neural-fragility" (a measure that indicates a node's susceptibility to destabilizing a network). The major reported findings included a correlation of amyloid-beta with higher neural-fragility and higher neural-excitability, and tau with higher neural-excitability. The epileptiform phenotype of AD was associated with higher neural-fragility, but not higher neural-excitability. The authors conclude that "diverse mechanisms" of E/I imbalance are associated with AD pathology and the epileptic phenotype.

Overall, the manuscript is of high quality and addresses an important topic in AD. The group effects comparing the aperiodic slope and fragility measures between HC and AD (Fig 2) were robust and demonstrate a clear group difference, with higher values of both measures in AD. The association of these measures with AD pathology was also a compelling demonstration of the differential association of amyloid and tau with measures of network physiology. However, several concerns are outlined below that should be addressed. In particular, the authors equate the aperiodic slope and fragility measures with E/I imbalance, but this has not been proven and is rather inferred from prior studies. More precise language and terminology should be used to refer to these measures in the methods, results, and discussion. In addition, the characterization of the AD-EPI+ group is lacking and limits the interpretation of these results. The authors do not actually measure amyloid or tau in this group, but inferences are made about the relationship between amyloid, tau and the epileptiform phenotype in the discussion that are not well-supported by the data. Detailed comments to address these, and other issues are outlined further below.

Methods

1. The methods state that there are 50 participants with M/EEG (30 epi+/20 epi-), but text and supplement indicate there are 47 (27 epi+/20 epi-). Please clarify.

2. Re: AD-EPI cohort: This group needs to be better described. The authors state "Clinical and other electrophysiological characteristics of this sub-cohort has been presented in earlier reports" (Methods, Pg 16, line 21), but no references are provided. Have all of these pts been reported in previous studies? If so, which studies? What were the characteristics of their epileptiform abnormalities (location, frequency, etc?). How many were taking anti-seizure medications?

3. The authors should provide details on how much time was elapsed between M/EEG, MRI, PIB PET, Tau PET, and FDG-PET studies. Did they set a time limit for how far apart these studies could be from one another, in order to be included in the analysis? To make these correlations, it would be prudent to limit the time between all studies to no more than 1 year for a given participant.

4. MEG used a 60 second segment of awake, eyes closed for analysis for each patient. Why use such a short amount of recording, if the MEG studies were 1 hour long? How noisy are these E/I measures depending on sample time selection?

5. Neural fragility measure -- I cannot comment on the technical aspects of these measurements, would suggest a reviewer who was involved in developing this method.

6. Fig 3B-F and Fig 4D-G. In these analyses, each subject is represented 210 times (210 cortical regions for each subject) and it is likely that the data from one cortical region would be co-linear with nearby regions in terms of their association with the outcome variable (e.g. fragility). Is this accounted for statistically? If this analysis was performed for only a single cortical region of interest in each, would the same association be found?

Results

7. Neuronal excitability and fragility in AD: It is not clear why z-scores vs. raw scores for neural excitability and neural fragility are used in different models? Raw values of excitability and fragility are used for the excitability vs. fragility correlation, and for AD vs. control group differences, but Z-scores of excitability and fragility are used for correlation with PET studies, and for the AD-EPI+ vs EPI- group differences. Can the authors clarify this?

8. The assumption that the aperiodic activity in the power spectrum inherently represents local "neural hyperexcitability" from intrinsic neuronal synaptic processes may be a stretch. The cited papers do not prove this to be the case. The same is true for neural fragility representing long-range synaptic input integration and regional E/I imbalance triggered by distant synaptic inputs. The language in the manuscript conflates what is actually measured with what the authors' interpretations of those measures are. I suggest that the authors use more objective terms in the reporting of these measures in the methods and results, and then discuss their interpretation of these results and the implications for E/I imbalance separately.

9. The author's refer to 18F-FDG PET as a quantification of neurodegeneration. While hypometabolism on FDG-PET is an accepted measure of neurodegeneration as part of the A/T/N criteria for AD, its interpretation is quite different in epilepsy. I suggest that the authors report the FDG-PET findings as hypometabolism rather than neurodegeneration, to avoid over-interpretation of physiological implications.

10. FDG-PET analyses: The authors z-scored the FDG-PET data to a "normative" dataset in a group of adults that was significantly older than the AD group (73 years old vs 62 years old). Why was this done (compared to the amyloid and tau PET data, which was analyzed as SUVR values?). What are the implications of doing this z-score, and could this potentially account for the findings in Figure 4F? Do the results still hold if the authors use SUVR values without z-scoring?

11. AD-EPI+ vs AD-EPI- analysis:

- How many of these participants were on anti-seizure medications, and was this controlled for in the analyses?
- There were no analyses done looking at amyloid, tau, and FDG in this group. Are there differences in amyloid and tau burden between these groups? Given that amyloid, tau and hypometabolism were correlated with E:I measures, shouldn't these be included in the model to control for these effects?
- Given that tau is necessary for amyloid-related hyperexcitability, was there any correlation between an amyloid*tau interaction and these measures of E:I imbalance?
- The authors do not provide any information from the ADI-EPI+ group in terms of where epileptiform activity occurs or how frequently this occurs. Note that the neural fragility measure was developed to identify very focal regions of the seizure onset zone in patients with epilepsy. Did the authors examine neural fragility specifically in regions of epileptiform activity in the AD-EPI+ group? Along similar lines, have the authors examined whether neural fragility is associated with the burden of focal epileptiform activity in this group?
- Is it concerning that the measure of "neural excitability" (slope of aperiodic activity) is not different in AD-EPI+ vs EPI-? What is the aperiodic slope measuring in this case?

12. All participants all underwent cognitive battery as indicated in Table S1, but this data was not analyzed otherwise. Did the E/I or fragility measures correlate with measures of cognitive performance?

Discussion

13. Related to #11 above. The finding that the measure of "neural excitability" was not increased (if anything the trend is the opposite) in the AD-EPI+ vs AD-EPI- groups, calls into question the suggested label "neural excitability" and representation of this measure as an indicator of local hyperexcitability or increased E/I imbalance. The presence of epileptiform activity would presume a higher E/I imbalance, at least in the region that the epileptiform activity is observed. This discordance is not specifically discussed or explained.

14. "The differential associations in the current results— tau distinctly correlated with local synaptic deficits and A β showing positive relations to both local and long-range synaptic integration deficits and support the diversity of tau and A β associated mechanisms of E/I imbalance". This is overstated in terms of interpreting the physiological implications of what was measured.

15. In discussing the "epileptic phenotype of AD" - I presume that the AD-EPI+ participants mostly had subclinical epileptiform activity (based on prior publications from this group), but this is not well-described in the current paper. I think it is important to distinguish between AD patients with subclinical epileptiform activity and those with clear clinical epileptic seizures in the discussion. The underlying pathologic mechanisms may not necessarily be the same, and conflating the two in the discussion may undermine the authors' point regarding the "diversity" of E/I imbalance that occurs in AD.

16. "The current finding that AD-EPI+ having higher neural fragility than AD-EPI-, despite similarly increased neural excitability, suggest that epileptic events are not a uniform manifestation but rather represent an AD phenotype with an added burden of E/I imbalance. As neural fragility is only correlated to A β , it is likely that this greater disease burden is contributed by A β vulnerable mechanisms—a phenomenon supported by both animal and clinical data."

-The phrase "added burden of E/I imbalance" is confusing here, as the authors previously use E/I imbalance as a term that encompasses both neural excitability and neural fragility.

-Just because the measure of neural fragility was associated with AB, and neural fragility was higher in AD-EPI+, does not necessarily mean that AB is what drives the epileptic phenotype.

17. The authors did not directly examine amyloid or tau pathology in the AD-EPI+ vs EPI- group comparisons. Why not? This should be stated as a major limitation.

18. "Second, clinical studies in AD patients also suggest that seizures are more likely to be associated with amyloid pathology rather than tau pathology. Seizures and epileptic manifestations are more common in patients with early-onset AD and during the initial stages of the disease likely reflecting the association between increased seizure activity and higher A β burden."

- I disagree with these statements and think a more balanced discussion is needed. If seizures in AD were more likely associated with amyloid pathology, then they would be seen more commonly in pre-clinical stages of AD (when amyloid burden is high, without significant tau accumulation). While prior studies have shown that seizures can occur in preclinical stages of AD, this does not happen in the majority of cases (Vossel et al, JAMA Neurology 2013; Sarkis et al, J Neuropsychiatry Clin Neurosci 2016), and many seizures happen at or after onset of clinical symptoms, which is more closely correlated with accumulation of tau pathology. Moreover, see DeVulder et al, Brain 2025, Lam et al, Neurology 2025, which demonstrate associations between local tau pathology and epileptiform activity and seizures in AD.

19. "The finding that A β pathology likely drives epileptic activity in the AD-EPI+ phenotype has important consequences for research focused on understanding how A β affects E/I imbalance."

- The analyses done in this study do not lead to the conclusion that "A β pathology likely drives epileptic activity in the AD-EPI+ phenotype". There was no analysis of amyloid or tau done in the AD-EPI+ group to directly support this statement.

20. Section on "Implications on protein lowering therapeutic trials and novel targets in clinical trials"

- I think this section should be shortened significantly. While I agree that these measures of E:I imbalance could potentially be interesting functional biomarkers to assess therapeutic effects in AD, it is not clear to me that these will necessarily be useful. Some of this discussion (regarding CAA and anti-seizure therapies) is not directly relevant to the findings of this paper.

Limitations:

21. Should include that this is an early onset AD cohort, with mean age of 62, and even lower for the AD-EPI+ group (60) -- which means onset of clinical symptoms may have been in the late 50s, and it is not clear whether these results would hold in a more typical AD cohort.

Conclusions:

22. "Whereas tau distinctly affects local excitability processes, A β affects both local and long-range synaptic input integration processes, with a stronger influence on the latter."

- Just because tau did not correlate with neural fragility, does not necessarily mean that it only affects local excitability processes. Neural fragility is just one measure of the larger network.

23. "Importantly, AD-EPI+ phenotype represents an added burden of E/I imbalance specifically related to A β associated long-range integration deficits."

- This conclusion is not totally clear to me.

Reviewer #3

(Remarks to the Author)

I co-reviewed this manuscript with one of the reviewers who provided the listed reports. This is part of the Nature

Communications initiative to facilitate training in peer review and to provide appropriate recognition for Early Career Researchers who co-review manuscripts.

Reviewer #4

(Remarks to the Author)

This is an interesting study dealing with a topic of significant value in the attempt of understanding dementia pathophysiology, in particular for the Alzheimer's Disease type. Methodology is advanced and sound enough and results/conclusions might be of general interest when reviewed under the comments/criticisms I will try to share with the Authors.

A major comment/criticisms is on the general topic, namely the changes in excitability of the AD brain and the linkage it might have with the increased rate of epilepsy with respect to non demented age-gender-education matched elderly population and with the neurodegenerative processes. Is this a specific marker of the demented brain? We know that in the 'natural epidemiology' of primary epilepsies there is a peak defined late onset epilepsy in elderly population. We also know that the risk of epilepsy is increased following an acute brain lesion like in stroke and in slowly growing lesion like for a brain neoplasm. What do have such conditions in common between them and AD? They have in common the acute/progressive loss of neurons, networks and connections and the brain reaction trying to resist to such a loss. One simple mechanism of brain resilience is the change in excitability in the attempt of recruiting 'silent' synapses/circuits to vicariate the function of the lost ones. In other words a simple and efficient mechanism which is active in several brain conditions from physiological brain aging to several and different brain pathologies; worth of mentioning, in all such conditions the rate of epilepsy is increased. On this basis how specific is what is described by the Authors within the frame of different 'specific markers' for AD? Moreover, there is a bulk of literature confirming what I just wrote utilizing Transcranial Magnetic Stimulation (TMS) and showing changes in brain excitability in the healthy elderly, in stroke, in AD and different forms of central nervous system neurodegeneration including amyotrophic lateral sclerosis. Authors are encouraged to consider such a comment and to include it in the 'limitations' section as well as in the introductory part of the text.

Other point –of less general impact on the experimental design- include:

1) Spiky EEG activity does not mean epilepsy since up to 10% of the general population can show them in the EEG without suffering any epileptic attack throughout their lives.. This aspect should be clarified. This observation tempers down the percentage of epilepsy of AD and dementias to about 10-15% of such population. Still higher than physiological and healthy brain, but less than reported in the text.

2) Mixing together early AD and MCI-due-AD might introduce a remarkable bias. It is in fact a common observation that in a MCI population only 30-to-50% progress to clinical dementia while the remaining is not even after a prolonged follow-up. This is also valid for MCI having positive biomarkers for beta/tau pathology even if at a lower level than those who have not. The reason for this is not clear, unless that the 'resilient' MCI amyloid/tau positive population must have protective factors which balance or block the activity of the risk factors. By putting together AD Patients and MCI subjects who might not develop dementia is introducing a confounding factor of importance in the final results.

3) The above point is strongly dealing with the debate between a 'biological' diagnosis and a 'clinical/neuropsychological diagnosis'. A consistent population of at-risk subjects (i.e. MCI population) is considered by the biological diagnosis supporters already affected by dementia when they carry positivity for given biomarkers even in absence of symptoms and progression and any significant impact in their living efficiency except for the tremendous burden due to a 'biological diagnosis' which will definitely change their social/professional/affective profile for several years until it is demonstrated that the 'biological diagnosis' was wrong in terms of a real disease.

Version 1:

Reviewer comments:

Reviewer #1

(Remarks to the Author)

All of my concerns have been addressed and discussed thoroughly, thank you. I have no further comments.

Reviewer #2

(Remarks to the Author)

The authors have addressed all of my major concerns, and I think the revised manuscript is greatly improved in terms of scientific strength and clarity. This is an important paper for the field!

Reviewer #3

(Remarks to the Author)

Reviewer #4

(Remarks to the Author)

Authors have addressed properly all my comments and suggestions. I feel that this manuscript might be now ready for publication

Response to Reviewers' Comments

We thank the referees for reviewing our manuscript and providing valuable feedback. We are extremely pleased that reviewers found several strengths and laudable aspects in our study and also made several important and helpful suggestions to improve the manuscript. We have incorporated all of the suggestions in this revision. What follows is a detailed point-by-point response to all the points raised by each reviewer. The revisions are marked in yellow-highlight in the revised manuscript.

The page-numbers and paragraph numbers mentioned at each response below refer to the PDF document of the revised manuscript which indicates the revisions in yellow highlight.

Reviewer 1

Ranasinghe and colleagues are reporting on magnetoencephalography (MEG) differences between Alzheimer's disease patients and controls and the association with 1) proteinopathies (tau and amyloid) and neurodegeneration and 2) epileptiform activity. The study claims that their approach helps to understand the how Excitation/Inhibition (E/I) balance is altered across the brain and its distinct associations with amyloid, tau and neurodegeneration, and whether patients with epileptiform activity present a distinct group of patients. In general, the need for better understanding the role of E/I imbalance in AD patients and its relation with amyloid, tau and neurodegeneration in human AD is high. The use of MEG and PET scanning to link large-scale neuronal function to underlying proteinopathies is a promising approach, and the author group possesses substantial expertise in this field. The methods used are innovative and a subset of the cohort consists of unique multimodal imaging data, including Amyloid, tau, FDG- PET and MEG. The finding that 1) amyloid and tau present distinct associations with E/I balance and 2) AD patients with epileptiform activity present a distinct group of patients are highly relevant for future therapeutic implications. The structure, clarity and context of the manuscript is of high quality. The sample size of this study appears adequate, the cohorts are well-characterized, and the conclusions are supported by results. I therefore consider the aim and the scope of this paper suitable for publication in Nature Communications. I only would like to share minor remarks which I would like to ask the authors to address before being able to recommend the manuscript for publication.

1) *General remarks*

- a) Overall, local excitability and interregional synaptic integration are presented as two distinct, complementary mechanisms throughout the paper. This is also supported by a lack of correlation between the measures in control subjects and the distinct spatial distributions over the cortex. However, local excitability is also greatly dependent on synaptic inputs from other regions, and, therefore, I think one should be careful in presenting local excitability and interregional synaptic integration as two distinct entities. Please, could the authors reflect upon this matter and, if they agree, nuance these statements throughout the manuscript?

We completely agree with the reviewer that the local excitability of a neuronal ensemble is influenced by both intrinsic neuronal properties and interregional synaptic integration. Indeed, the spectral aperiodic slope, which we have used as a proxy for 'neural excitability', likely reflects contributions from both local and long-range synaptic processes. In contrast, neural fragility, as originally described by Sritharan and Sarma (2014)¹, quantifies linear dynamic instability within a local region that is explicitly modulated by long-range synaptic inputs. As the reviewer correctly notes, however, the observed lack of correlation between the spectral aperiodic slope and neural fragility in control subjects, along with their distinct spatial distributions, suggests that these two metrics capture partially independent features of neurophysiological processes. While the aperiodic slope is influenced by synaptic inputs, our data indicate that in the context of human neuroimaging, it is predominantly shaped by local synaptic and neuronal mechanisms. We have revised the manuscript to clarify the nuanced relationship between these two measures. In the introduction, we now explicitly define 'neural excitability' as reflecting "local intrinsic neuronal and synaptic processes" while acknowledging its susceptibility to extrinsic modulation. Additionally, in the results section when we introduce both metrics, we have included the following clarification: "The two metrics, although conceptually distinct, capture complementary aspects of excitation/inhibition (E/I) dynamics. Importantly, while neural excitability primarily reflects intrinsic local activity, it also incorporates contributions from synaptic inputs. In contrast, neural fragility explicitly models the impact of long-

range synaptic input integration on E/I balance.” We thank the reviewer for highlighting this critical distinction, which has strengthened the interpretive framework of our study. (page 5, line 18-20; page 6, lines 9-13)

- b) Furthermore, the authors correctly chose the aperiodic activity as a measure of local neuronal excitability. However, it would be fair to introduce that various potential quantitative measures of E/I exist for M/EEG [for example, see Ahmad et al. 2022 Transl Psych]. Most of these (including 1/f [A systematic review of aperiodic neural activity in clinical investigations | medRxiv]) are at their infancy, requiring further research to elaborate upon their robustness and validity. Furthermore, the use of 1/f as measure of excitability is not straightforward [Brake et al 2023 Nat Comm], which is supported by the findings that 1/f does not associated with epileptiform activity in the current study. The limitations of the chosen measures to quantify E/I should at least be addressed in the discussion section.

We thank the reviewer for the comment. We fully agree that the aperiodic slope, while already shown in multiple studies to be a sensitive and reliable indicator of local neuronal excitability, remains an evolving tool with respect to its optimal technical implementation and interpretability. As the reviewer correctly noted, a key finding in the current study indicates that although the aperiodic slope robustly reflects local synaptic and neuronal processes contributing to excitation/inhibition (E/I) dynamics, it does not necessarily indicate the specific E/I dysfunction associated with epileptic manifestations in AD. This observation does not reflect a limitation of the metric per se, but rather highlights the complexity and heterogeneity of E/I alterations in AD. We have revised the discussion to address this important point, incorporating the relevant citations as suggested. Furthermore, we emphasize the importance of developing standardized methodologies for spectral slope analysis to enhance consistency across studies and accelerate progress in this promising area of research. (Discussion: page 12, line 23; page 13 lines; Limitations: 1-2; page 19, lines 12-17)

2) Introduction

- a) The authors conclude in the first paragraph that none of the AD mouse models capture the full complexity of AD dual proteinopathy, which I think is true. However, there have been several approaches aiming to model both amyloid and tau pathology in mice, which deserve some attention (for instance: Tok, Crauwels and Drinkenburg (2022) Journal of Neurology and Experimental Neuroscience.).

Thank you for the comment. We acknowledge this work in the revised manuscript. (page 3, lines 15-16)

- b) I have some conceptual remarks about the use of the words vulnerability and heterogeneity. The authors state that AD patients with epileptiform activity present ‘additional disease vulnerabilities’ and hypothesize that these subjects have ‘greater vulnerabilities of E/I balance’, but it is not clear to me what is meant with vulnerability here. Does it mean that the same load of amyloid or tau pathology elicits more severe E/I imbalance (resulting in epileptiform abnormalities) compared to other patients? If so, what would be potential mechanisms for this increased vulnerability? Or does it mean that similar E/I values can have distinct manifestations (if so why)? Please, could the authors elaborate upon this?

We agree with the reviewer that the term vulnerability is confusing to the reader and our study do not specifically quantify vulnerability. In the revised manuscript, we have removed the term vulnerability and explicitly mention the hypothesis statement as predicting “AD-EPI+ cohort will represent a distinct phenotype of E/I imbalance in AD”. (page 5, lines 8-10)

- c) One of the provided explanations for why only a subset of patients show epileptiform activity is “heterogeneity”. The authors subsequently explain a potential differential (i.e. heterogeneous) effect of amyloid and tau on E/I balance. Is this indeed the type of heterogeneity meant to explain differential presentation of E/I imbalance among AD patients or do the authors refer to a different kind of heterogeneity? (this also accounts for use of the word heterogeneity in discussion section). If the former is true, would it make more sense to compare the load of abeta and tau between EPI+ and EPI- subjects?

We completely agree with the reviewer’s opinion about two levels of heterogeneity. In the original manuscript the term ‘heterogeneity’ in the referenced sentence was used to indicate the differential

presentation of E/I imbalance, whereby only a subset of patients exhibits epileptiform activity. We now explicitly clarify this point in the revised Introduction addressing both types of heterogeneity the reviewer pointed out and discuss this in detail in the discussion section as well. (page 4, lines 7-15; page 12, under the subheading "Diverse manifestations of E/I imbalance in AD").

Quantifying amyloid and tau burden in AD-EPI+ versus AD-EPI- patients is indeed the logical next step and would provide important insights into the mechanistic associations between AD proteinopathy and E/I imbalance. However, the patients in the current sub-cohorts who underwent LTM-EEG and M/EEG protocols to define AD-EPI+ and AD-EPI- groups did not uniformly undergo amyloid- and tau-PET imaging. This limitation stems from the fact that LTM-EEG and M/EEG data collection is extremely resource-intensive and was conducted over a prolonged period beginning in 2013, whereas PET imaging protocols only became widely available for research use in more recent years. Nevertheless, we are currently conducting an ongoing longitudinal study that includes M/EEG, amyloid-PET, and tau-PET imaging in AD patients, which will allow us to address this question in future work. We have acknowledged this limitation in the revised 'Discussion'. (page 19, lines 7-10)

3) Methods

a) There is some missing information:

i) A statement that the study has been performed in accordance with the declaration of Helsinki.

We have added this information into the revised 'Methods'. (page 21, lines 18-19)

ii) The atlas used for source reconstruction of MEG data, and why the authors included only 210 cortical out of total of 246 ROIs.

Thank you for the question. We included the 210 regions representing the cortex and excluded the subcortex, as defined in the Brainnetome atlas², in our analyses. The decision to exclude subcortical regions was based on two main considerations: (1) AD neuropathology predominantly affects cortical regions, while subcortical structures are relatively spared; and (2) MEG source localization yields higher accuracy in cortical regions compared to subcortical areas. We have now included these additional details into the revised 'Methods'. (page 22, lines 17-19)

iii) The authors mention in the results section that for some statistical models CDR was included as covariate, but this is not reported in the methods, please check.

We thank the reviewer for pointing this out. We have revised the methods section which now includes these steps in detail. Furthermore, after constructive criticisms from other reviewers we have included additional analyses showing the effects of CDR in the LMM analyses and expanded the methods and supplementary results section with these details. (page 28, lines 5-6; page 29, lines 1-2; Supplementary figures: S1 and S3)

iv) Did the authors check assumptions for LMM (such as normal distribution, constant variance)? Did the authors expect a linear relationship between the measures?

We thank the reviewer for the question. In examining associations between measures of E/I imbalance and AD proteinopathy, we do not have specific hypothesis about the nature of the relationship (linear vs. non-linear). We selected linear mixed-effects models (LMM) because this approach minimizes bias and reduces the risk of overfitting. All statistical models used SAS Proc MIXED procedure and we assessed the distribution of residuals through histograms and evaluated skewness to confirm that the model assumptions were satisfied. Additionally, model convergence was verified for all analyses. We have included these additional methodological details into the revised manuscript. (page 27, lines 11-13)

b) A number of methodological choices require some additional argumentation or explanation:

i) Could the authors explain why for tau and amyloid the SUVR values are computed, but FDG PET is normalized to control data?

Thank you for this important question and allowing us to us to clarify our methodological approach. SUVR values for amyloid and tau represent the degree of abnormal protein deposition—amyloid plaques and hyperphosphorylated tau tangles, respectively. Therefore, the uptake patterns directly index the ‘abnormality’ without requiring normalization to a reference group. In contrast, FDG-PET measures cerebral glucose metabolism, which follows a well-characterized physiological distribution in healthy individuals. As such, abnormality in FDG-PET must be defined relative to normative patterns, which can be reliably computed from an age-matched healthy control population scans and estimated as ‘z-scores’. This normalization step was essential to account for the physiological spatial variability in our analyses designed to examine the relationships between E/I indices and abnormal metabolism in AD. These details have been added to the revised ‘Methods’. (page 27, lines 21-23; page 28, line 1)

- ii) Why are z-scores of E/I measures computed and used for some, but not all, statistical analyses?

All analyses in AD patients that examined the relationship between abnormal E/I and protein accumulation, or metabolic dysfunction used z-scored E/I measures. This normalization was necessary to account for the physiological spatial variability of E/I observed in the healthy human brain. In contrast, analyses assessing group differences in E/I were not affected by this variability, as the group comparison approach inherently accounts for it. We have clarified the rationale for z-score computation and its application in the revised Methods section. (page 27, lines 13-16)

- iii) What is an age-corrected unpaired t-test (line 6 p 23)? Do the authors mean ANCOVA?

Thank you for the comment. We have now corrected the terminology as Analysis of Covariate models in the revised manuscript. Specifically, we performed covariate adjustment for age when examining group differences between AD patients and controls on neural metrics. This approach corresponds to an ANCOVA framework, which falls under the umbrella of the General Linear Models. It evaluates the effect of a categorical independent variable on a continuous dependent variable while accounting for the influence of one or more continuous covariates. The resulting estimate of group effect is a t-value reflecting group difference adjusted for age. (page 29, lines 20-22)

- c) Some other thoughts:

- i) Should CDR or another indicator of ‘disease stage/duration’ be considered as covariate for all models (do the authors expect similar correlations for MCI vs probable AD stage?)

We thank the reviewer for the suggestion. We have now re-run all statistical models including CDR as a categorical covariate. These results are presented in the Supplementary Figures and referenced in the Results section. In brief, inclusion of CDR did not alter the main findings from the linear mixed-effects models, and no significant differences were observed between the slopes for CDR = 0.5 and CDR = 1. (Please also see Reviewer #3, Comment 2). (page 28, lines 5-6; Supplementary figures: S1 and S3)

- ii) Why did the authors choose to estimate the association between E/I and neurodegeneration after correcting for abeta/tau and not, for example, estimating the association between E/I and tau when correcting for amyloid? I think the latter would have given additional evidence about the specificity of tau with E/I manifestations, independent of amyloid.

We thank the reviewer for the comment. We have now conducted additional analyses estimating the association between E/I and tau pathology while controlling for amyloid burden. In the subset of patients with tau-PET imaging (n = 35), who underwent amyloid imaging with different tracers, we included global amyloid burden expressed in centiloid units as a covariate. In a separate analysis restricted to the subset evaluated with tau-PET and with PIB-PET as the uniform tracer (n = 23), we included regional amyloid as a covariate. These analyses revealed that the association between neural excitability and tau pathology is independent of amyloid accumulation. The results are now included in the revised Supplement and referenced in the revised ‘Results’ section and methods included in the revised ‘Methods’ (Results: page 9, lines 1-9; Methods: page 28, lines 6-13; Supplementary figure: S2)

- iii) One may expect that the locations showing distinct fragility will overlap with epileptiform activity location, but this is not reported upon. I believe this data should be available from previous reports and including this in the manuscript would enhance the credibility of the findings.

Thank you for this insightful comment. Examining the regional distribution of epileptiform activity and its potential overlap with neural fragility is indeed a critical question. However, the current dataset does not include systematically localized epileptiform events, as the AD-EPI+ classification was based solely on clinical identification of spikes or sharp waves without detailed spatial mapping. A regional analysis of epileptiform events in AD-EPI+ is indeed an important study itself, which however necessitates spike localization and dipole fitting for individual epileptic events and generating maps of seizure semiology for each AD-EPI+ patient. While this extensive analysis is beyond the scope of the current manuscript, we are pleased to mention that this is one of our ongoing projects and we expect to publish the findings in a separate manuscript in the future. We acknowledge this in the revised manuscript. (page 19, lines 10-12).

4) Discussion

- a) It seems as if the authors tend to infer a direction of effect between E/I measures and PET (“E/I imbalance is driven by/stemming from .. PET”), but considering the cross-sectional nature of the data, one should be careful making such statements. I would suggest to nuance these statements a bit to avoid over interpretation of results.

Thank you. We have revised the manuscript and removed the term “stemming from” accordingly.

- b) The authors report distinct neural fragility between EPI+ and EPI- AD patients in specific locations, but similar local neural excitability. Do these findings suggest 1/f is not an appropriate measure for E/I?

Thank you for the question and allowing us to explain further. The current manuscript demonstrates that the E/I imbalance in AD is diverse, involving both altered long-range synaptic integration and local neuronal/synaptic deficits. What our results indicate is that 1/f is certainly a sensitive measure of altered E/I associated with local neuronal and synaptic deficits in AD but does not specifically capture the E/I abnormalities associated with epileptiform activity in AD patients. In contrast, neural fragility more accurately reflects the E/I abnormalities associated with epileptiform activity in AD patients. (We have clarified this in the discussion section under the subheading “Diverse manifestations of E/I imbalance in AD”)

- c) In general, I believe the discussion would benefit from inclusion of a short paragraph comparing results to previous clinical MEG reports and combined MEG/PET studies in AD, since this is not the first paper studying the relation between E/I in AD and controls or MEG in relation to abeta/tau. [Wiesman et al 2022 Brain; van Nifterick et al 2023 SciRep; Martinez-Canada et al 2023; Gallego-Rudolf et al 2024 NatComm; Schoonhoven et al., 2023 Brain; Javed et al 2025 JoN]. Also, a key finding of this study is a strong correlation between local hyperexcitability and neurodegeneration; How does this relate to previous literature?

Thank you for the suggestion. We have added a new section titled “E/I imbalance as a core element of neural circuit dysfunction” in the revised Discussion. This section contextualizes our findings with previous MEG and combined MEG/PET studies in AD and discusses how our results, including the correlation between local hyperexcitability and neurodegeneration, and have added relevant citations. (page 17-18; page 16, lines 2-4).

- d) Please, provide some words on the limitations of used metrics for E/I in the discussion section (as mentioned before)

We have revised the ‘Limitations’ section including these details. Please also see the response to the comment under general remarks-(b). (page 19, lines 12-20)

Reviewer #2

The authors aim to quantify excitatory-inhibitory (E/I) imbalance in AD using MEG and subsequently correlate these measures with AD pathology and epileptic activity. They examine two E/I, termed “neural-excitability” (aperiodic slope of the power spectrum) and “neural-fragility” (a measure that indicates a node’s susceptibility to destabilizing a network). The major reported findings included a correlation of amyloid-beta with higher neural-fragility and higher neural-excitability, and tau with higher neural-excitability. The epileptiform phenotype of AD was associated with higher neural-fragility, but not higher neural-excitability. The authors conclude that “diverse mechanisms” of E/I imbalance are associated with AD pathology and the epileptic phenotype.

Overall, the manuscript is of high quality and addresses an important topic in AD. The group effects comparing the aperiodic slope and fragility measures between HC and AD (Fig 2) were robust and demonstrate a clear group difference, with higher values of both measures in AD. The association of these measures with AD pathology was also a compelling demonstration of the differential association of amyloid and tau with measures of network physiology. However, several concerns are outlined below that should be addressed. In particular, the authors equate the aperiodic slope and fragility measures with E/I imbalance, but this has not been proven and is rather inferred from prior studies. More precise language and terminology should be used to refer to these measures in the methods, results, and discussion. In addition, the characterization of the AD-EPI+ group is lacking and limits the interpretation of these results. The authors do not actually measure amyloid or tau in this group, but inferences are made about the relationship between amyloid, tau and the epileptiform phenotype in the discussion that are not well-supported by the data. Detailed comments to address these, and other issues are outlined further below.

We thank the reviewer for taking time to critique our manuscript. Each of the points that reviewer mentioned in the summary paragraph have been addressed under the detailed point-by-point comments below. For easy reference we cite here which items address these main points from the list below.

1. More precise language and terminology to refer to E/I measures in the methods, results, and discussion: ***This is addressed under #8***
2. Characterization of the AD-EPI+ group: ***This is addressed under # 2 and #11***
3. The authors do not actually measure amyloid or tau in this group, but inferences are made about the relationship between amyloid, tau and the epileptiform phenotype in the discussion. ***This is addressed under # 14 and 17)***

Methods

- 1) The methods state that there are 50 participants with M/EEG (30 epi+/20 epi-), but text and supplement indicate there are 47 (27 epi+/20 epi-). Please clarify.

Thank you for pointing out this oversight. We have corrected this typo in the revised manuscript. The current investigation included 47 AD patients including 27 AD-EPI+ and 20 AD-EPI-. (page 24, lines 2-3)

- 2) Re: AD-EPI cohort: This group needs to be better described. The authors state “Clinical and other electrophysiological characteristics of this sub-cohort has been presented in earlier reports” (Methods, Pg 16, line 21), but no references are provided. Have all of these pts been reported in previous studies? If so, which studies? What were the characteristics of their epileptiform abnormalities (location, frequency, etc?). How many were taking anti-seizure medications?

Thank you for the comment. We have now included additional details on the detection of subclinical epileptiform activity and the identification of the AD-EPI+ cohort in the Methods section. We explicitly reference the prior publication (Ranasinghe et al., Brain, 2022)³, which reported the lobar-level distribution of epileptiform activity in this cohort. None of the patients were taking anti-seizure medications. Clinical and demographic characteristics of the AD-EPI+ and AD-EPI- sub-cohorts are presented in Supplementary Table S3 (page 21, line 5-11; Supplementary table S3)

- 3) The authors should provide details on how much time was elapsed between M/EEG, MRI, PIB PET, Tau PET, and FDG-PET studies. Did they set a time limit for how far apart these studies could be from one another, in order to be included in the analysis? To make these correlations, it would be prudent to limit the time between all studies to no more than 1 year for a given participant.

All MRI scans used in the quantitative analyses were acquired within 12 months of the MEG scan (mean \pm SD time difference: 0.18 ± 0.40 months). This information has been added to the Methods section (page 26, lines 12-13).

In addition, we have included the details of time differences between MEG and PET into the revised supplement (*Supplementary table S2*), which includes the following details: 46 out of 52 AD patients imaged with ¹¹C-PIB were imaged within 12 months from the MEG scan and the rest of the 6 patients were imaged within 32 months; 33 out of 35 AD patients imaged with flortaucipir were imaged within 12 months from the MEG scan and the remaining 2 patients were scanned within 24 months. 43 out of 51 AD patients scanned with FDG-PET were imaged within 12 months from the MEG scan; 7 of the remaining 8 patients were scanned within 32 months; one patient was scanned within 55 months from the MEG scan.

We repeated the LMM examining the associations between FDG-PET and MEG derived neural excitability and neural fragility metrics (depicted in Fig.4D-E), after excluding the subject who had the time difference of 55 months between MEG and FDG-PET and the findings remained the same (Neural excitability: $F=18.90$, $P<0.0001$; Neural fragility: $F=0.37$, $P=0.5447$). Therefore, this subject was retained in the analyses.

All statistical models included the time difference between MEG and relevant PET scan as a covariate. We strongly believe that inclusion of all data—particularly since only a small proportion of participants in each cohort had imaging beyond the 12-month window—enhances the statistical power of our analyses and supports more reliable and accurate interpretations.

- 4) MEG used a 60 second segment of awake, eyes closed for analysis for each patient. Why use such a short amount of recording, if the MEG studies were 1 hour long? How noisy are these E/I measures depending on sample time selection?

We thank the reviewer for the comment and the opportunity to provide a detailed explanation. A 60-second MEG segment sampled at 600 Hz yields an extremely rich dataset for examining neural activity patterns in the resting human brain. Both our group and others have demonstrated that a 60-second epoch of continuous brain activity provides a reliable time window with minimal artifacts and noise^{4, 5, 6, 7}. This has been validated across various populations, including young and older adults, and in both healthy individuals and patients with neurological conditions such as Alzheimer's disease (AD), epilepsy, brain tumors, and schizophrenia^{5, 8, 9}. The one-hour MEG/EEG recording protocol is the current diagnostic guideline for epilepsy in clinical readouts from M/EEG and includes an initial awake resting period, followed by a period during which participants are permitted to fall asleep¹⁰. This approach facilitates the detection of subclinical epileptiform activity, which may emerge during drowsiness or sleep. For our quantitative electrophysiological analyses, we selected a 60-second segment from the initial awake, eyes-closed resting state. We have excluded the analysis of the M/EEG data during the drowsy/sleep period, which is a separate ongoing investigation by our group. The selected 60 second data epoch was confirmed to be of resting awake condition and to be free of artifacts.

The E/I measures exhibit limited sensitivity to sample time selection as these measures are based on steady-state properties rather than rapidly fluctuating dynamics. For neural excitability, we use 1 min of data, which comprises 36000 samples to derive the steady state power spectrum and compute the aperiodic slope for each brain region. For neural fragility, we use the same data sizes to estimate linear dynamics model for the whole brain network and calculate stability measures for perturbation of each brain region. Given that both measures target steady-state characteristics, the sampling is fully sufficient to ensure robust and reliable estimation.

- 5) Neural fragility measure -- I cannot comment on the technical aspects of these measurements, would suggest a reviewer who was involved in developing this method.

Thank you.

- 6) Fig 3B-F and Fig 4D-G. In these analyses, each subject is represented 210 times (210 cortical regions for each subject) and it is likely that the data from one cortical region would be co-linear with nearby regions in terms of their association with the outcome variable (e.g. fragility). Is this accounted for statistically? If this analysis was performed for only a single cortical region of interest in each, would the same association be found?

Figures 3B–F and 4D–G present the results of linear mixed model (LMM) analyses. As the reviewer correctly noted, these models incorporated a repeated measures design to account for the 210 brain regions assessed per subject. Specifically, the models included both random intercepts and random slopes to model subject-specific variation. In repeated measures data, observations within the same

individual are inherently correlated; modeling random effects allows LMMs to appropriately account for this intra-subject co-linearity, thereby avoiding inflated Type I error rates that would arise from treating these observations as independent. Furthermore, LMMs offer flexible covariance structures that can be tailored to the specific correlation pattern among repeated observations, enabling a more accurate representation of within-subject variability. By partitioning within- and between-subject effects, LMMs reduce the risk of biased fixed-effect estimates caused by collinearity. This modeling framework enabled us to analyze the complete whole-brain multimodal imaging data in a statistically rigorous and unbiased manner, avoiding the selection bias that will be introduced by an *a priori* region-of-interest decisions. While a similar association might be observed in a single cortical region of interest (ROI), this would depend on the specific region selected and its relevance to the underlying biological process. However, focusing on a single ROI introduces substantial limitations. It assumes that the effect is localized and risks both selection bias and reduced generalizability. Our approach, using a whole-brain analysis within a linear mixed model framework, allows us to capture distributed effects across multiple regions without making a priori assumptions. This design increases statistical power and ensures that associations are not driven by arbitrary region selection. Post hoc inspection of specific ROIs based on the whole-brain results, however, could be used in follow-up analyses to explore localized effects in more detail. We have now included details of the rationale of our approach into the revised methods. (page 27, lines 5-13)

Results

- 7) Neuronal excitability and fragility in AD: It is not clear why z-scores vs. raw scores for neural excitability and neural fragility are used in different models? Raw values of excitability and fragility are used for the excitability vs. fragility correlation, and for AD vs. control group differences, but Z-scores of excitability and fragility are used for correlation with PET studies, and for the AD-EPI+ vs EPI- group differences. Can the authors clarify this?

Thank you for the comment. We now clarify this point in the revised Methods: z-scores were used in analyses examining relationships between E/I and molecular or metabolic abnormalities to account for normal regional variability in E/I across the cortex. In contrast, raw E/I values were used in group comparisons (e.g., AD vs. controls), where regional variability is inherently controlled by the comparison itself. (page 27, lines 13-16 and lines 21-23)

- 8) The assumption that the aperiodic activity in the power spectrum inherently represents local “neural hyperexcitability” from intrinsic neuronal synaptic processes may be a stretch. The cited papers do not prove this to be the case. The same is true for neural fragility representing long-range synaptic input integration and regional E/I imbalance triggered by distant synaptic inputs. The language in the manuscript conflates what is actually measured with what the authors’ interpretations of those measures are. I suggest that the authors use more objective terms in the reporting of these measures in the methods and results, and then discuss their interpretation of these results and the implications for E/I imbalance separately.

We thank the reviewer for bringing up this point for further clarification.

Regarding neural fragility, we refer to the original work by Dr. Sridevi Sarma and colleagues^{1, 11, 12}, who introduced the terminology ‘neural fragility’ and the specific metric to estimate long-range synaptic integration deficits contributing to regional E/I imbalance. Our study directly adopts the methodological framework established by these investigators, and the term neural fragility is used in accordance with its original definition.

Regarding neural excitability, as correctly noted by the reviewer, this term is used in our study to refer to the aperiodic slope of the power spectrum. This reference was to support the reader with the findings presented in parallel to ‘neural fragility’. While this measure does not excitability capture the local excitability in all its biophysical forms, a substantial body of literature has demonstrated that spectral aperiodic component is a reliable proxy for the shifts in local excitation-inhibition dynamics contributed by neuronal and synaptic processes^{13, 14}. Recent reports synthesizing these biophysical properties further demonstrate that a flatter aperiodic slope reliably capture increased cortical excitability, although the precise mechanistic substrates are still under active investigation^{15, 16, 17}. We have clarified these details in the revised introduction with relevant citations. (page 4, line 18-23; page 6, lines 9-13; please also see the response to Reviewer #1 - General remarks: a)

Having said the afore mentioned explanations, we agree with the reviewer that more objective terminology will help the reader to relate these findings in accurate context. In the revised manuscript, we have explicitly clarified that the terms neural excitability and neural fragility refer to the aperiodic spectral slope and linear dynamic instability, respectively. These descriptors now accompany every mention of the metrics in the figures presenting our methods and findings (All figures are revised including the objective

descriptions of neural excitability and neural fragility as aperiodic spectral slope and linear dynamic instability, respectively; page 5, lines 18-20; page 6, line 2-3)

- 9) The author's refer to 18F-FDG PET as a quantification of neurodegeneration. While hypometabolism on FDG-PET is an accepted measure of neurodegeneration as part of the A/T/N criteria for AD, its interpretation is quite different in epilepsy. I suggest that the authors report the FDG-PET findings as hypometabolism rather than neurodegeneration, to avoid over-interpretation of physiological implications.

Thank you. The revised manuscript is using the terminology 'hypometabolism' instead of 'neurodegeneration' when referring to the effects of FDG-PET.

- 10) FDG-PET analyses: The authors z-scored the FDG-PET data to a "normative" dataset in a group of adults that was significantly older than the AD group (73 years old vs 62 years old). Why was this done (compared to the amyloid and tau PET data, which was analyzed as SUVR values?). What are the implications of doing this z-score, and could this potentially account for the findings in Figure 4F? Do the results still hold if the authors use SUVR values without z-scoring?

We thank the reviewer for the comment. The rationale for z-scoring the FDG-PET data differs fundamentally from the use of SUVR values for amyloid and tau PET. Amyloid and tau PET SUVRs represent absolute measures of pathological accumulation (plaques and tangles, respectively), and thus their raw uptake values directly reflect disease-related abnormalities. In contrast, FDG-PET reflects regional glucose metabolism, which follows a well-established physiological pattern across the cortex. Raw FDG-PET SUVR values in patients with AD are therefore not inherently indicative of abnormality and to quantify metabolic abnormalities accounting for regional physiological variability, we estimate z-scores of the FDG-PET against a healthy control cohort. These details have been added to the revised 'Methods'. (page 27, lines 21-23)

We also wish to clarify that the control cohort used to compute z-scores was age-matched to the AD subgroup with FDG-PET data (mean age 61.36 ± 7.76 years for controls vs. 62.1 ± 6.9 years for AD patients). We have corrected this typographical error in the revised manuscript. (page 28, line 3).

- 11) AD-EPI+ vs AD-EPI- analysis:

- a) How many of these participants were on anti-seizure medications, and was this controlled for in the analyses?

None of the participants in the current study was on anti-epileptics. We have included this information into the revised methods. (page 21, lines 9-10)

- b) There were no analyses done looking at amyloid, tau, and FDG in this group. Are there differences in amyloid and tau burden between these groups? Given that amyloid, tau and hypometabolism were correlated with E:I measures, shouldn't these be included in the model to control for these effects?

We agree with the reviewer that quantification of amyloid and tau in AD-EPI+ vs. AD-EPI- patients is indeed the next ideal step in the analyses and would provide further insight into the potential mechanism proposed in the manuscript. However, we do not have the amyloid- and tau-PET imaging available for all the patients included in the current sub-cohorts which underwent LTM-EEG and M/EEG protocols to define the category identities of AD-EPI+ and AD-EPI-. This is because the AD-EPI protocol which has an extremely high demand for resources with overnight video monitoring of EEG followed by one-hour M/EEG, has been collected over many years starting from 2013 while the PET imaging modalities became available for research only within the last few years. Nonetheless, we currently have an ongoing longitudinal study in which we image AD patients using multimodal imaging of M/EEG, amyloid-PET and tau-PET, which will allow us to answer the question whether amyloid and tau are distinctly different in AD-EPI+ vs AD-EPI- as well as relate the E/I measures to proteins within each sub-cohort, and we will publish these findings in future manuscripts. We have addressed this limitation in the revised manuscript. See also (Reviewer #1: introduction (c)). (page 19, lines 7-12)

- c) Given that tau is necessary for amyloid-related hyperexcitability, was there any correlation between an amyloid*tau interaction and these measures of E:I imbalance?

We thank the reviewer for this insightful question. In transgenic mouse models of Alzheimer's disease, physiological tau has been shown to play a permissive role in mediating amyloid-related network hyperexcitability. Specifically, studies have demonstrated that reduction of endogenous tau levels can normalize hyperexcitability in mice overexpressing amyloid- β . However, current human neuroimaging modalities do not permit reliable quantification of regional physiological (non-pathological) tau. Available tau-PET tracers selectively bind to abnormally phosphorylated tau aggregates forming neurofibrillary tangles, rather than to soluble or physiological tau species. Consequently, our dataset—based on tau-PET imaging—does not allow for a direct assessment of the modulatory role of physiological tau on amyloid-related alterations in excitability. Thus, while the interaction between amyloid and tau pathology is a critical area of investigation, current in-vivo imaging limitations preclude us from directly investigating the enabling role of tau on amyloid related hyperexcitability this hypothesis within the current framework.

- d) The authors do not provide any information from the ADI-EPI+ group in terms of where epileptiform activity occurs or how frequently this occurs. Note that the neural fragility measure was developed to identify very focal regions of the seizure onset zone in patients with epilepsy. Did the authors examine neural fragility specifically in regions of epileptiform activity in the AD-EPI+ group? Along similar lines, have the authors examined whether neural fragility is associated with the burden of focal epileptiform activity in this group?

We thank the reviewer for this important point. (Please also see the related response to *Reviewer #1, Methods c.iii*). Regarding the question of whether we assessed neural fragility in relation to the burden of focal epileptiform activity, we would like to clarify that our AD cohort was dichotomized into EPI+ and EPI- groups based on clinical evidence of epileptiform abnormalities—defined by the presence of spikes or sharp waves on LTM- EEG and in M/EEG recordings. However, systematic quantification of the frequency or regional burden of these events was not performed. Specifically, epileptiform activity was not annotated in a time-resolved or spatially resolved manner, and the dataset does not include exhaustive marking of all events or their frequency per patient. Consequently, we did not perform regional analyses of neural fragility restricted to areas with documented epileptiform discharges. Such analyses would require individualized spike localization, dipole fitting for each identified event, and generation of spatial maps of epileptiform activity per subject—an effort that is methodologically extensive and beyond the scope of the current study. That said, we fully agree with the reviewer that such analyses are of high scientific value and directly relevant to understanding the relationship between network instability and regional epileptiform discharges in AD. We are currently pursuing this line of investigation in an ongoing study, and we plan to report these findings in a future manuscript. This limitation is acknowledged in the revised manuscript (*page 19, lines 7-12*)

- e) Is it concerning that the measure of “neural excitability” (slope of aperiodic activity) is not different in AD-EPI+ vs EPI-? What is the aperiodic slope measuring in this case?

We thank the reviewer for allowing use to explain further. It is not concerning that neural excitability (aperiodic spectral slope) is not different in EPI+ vs. EPI-. Our results clearly demonstrate that while all AD patients have increased neural excitability representing an E/I imbalance, AD patients who harbor subclinical epileptic manifestations, have a specific increase of neural fragility compared to those patients who do not have subclinical epileptic manifestations. We have further illustrated these findings, especially the fact that both AD-EPI+ and AD-EPI- have increased aperiodic slope compared to controls, in the revised manuscript (*Supplementary figure S5*).

- 12) All participants all underwent cognitive battery as indicated in Table S1, but this data was not analyzed otherwise. Did the E/I or fragility measures correlate with measures of cognitive performance?

Thank you for the suggestion. We have now included the associations between global cognition as measured by MMSE and neural excitability and fragility in the revised manuscript. These analyses demonstrated that both metrics were more abnormal in patients who showed greater degree of MMSE impairments. Given the current sample size, our approach of using global cognitive measures in these analyses increased the statistical power and minimized the floor/ceiling effects, compared to individual cognitive tests. (*Figure 2; page 7, lines 11-23; page 8, lines 1-3*)

Discussion

- 13) Related to #11 above. The finding that the measure of “neural excitability” was not increased (if anything the trend is the opposite) in the AD-EPI+ vs AD-EPI- groups, calls into question the suggested label “neural excitability” and representation of this measure as an indicator of local hyperexcitability or increased E/I imbalance. The presence of epileptiform activity would presume a higher E/I imbalance, at least in the region that the epileptiform activity is observed. This discordance is not specifically discussed or explained.

Thank you for the comment. A key finding from the current study is that that E/I imbalance in AD involves impairments at multiple levels along the functional architecture of neural circuits. The signature increase of neural fragility in AD-EPI+ compared to AD-EPI-, despite similarly elevated neural excitability, suggests that epileptic phenotype in AD represents a subset of patients who harbor greater impairments in long-range synaptic input integration processes. It is important note that all AD patients, however, exhibit E/I imbalance primarily contributed by local neural and synaptic dysfunctions. Collectively these findings provide neurophysiological basis for a seemingly contradictory observation from clinical investigations where studies using quantitative assays suggested a generalized increase of E/I in AD depicted by flattened aperiodic slopes, while several carefully constructed observational studies consistently found epileptic events only in a fraction of AD patients. The current results present an interpretation to these incongruent outcomes by demonstrating the heterogenous manifestations of E/I imbalance in AD where local hyperexcitability appears to be a more generalized feature, while specific deficits in long-range synaptic integration may underlie the epileptic phenotype. We have now clarified this in the revised discussion. (This is addressed under the discussion section under the subheading “Epileptic manifestations signify an AD phenotype with greater neural fragility” starting on page 13)

- 14) “The differential associations in the current results— tau distinctly correlated with local synaptic deficits and A β showing positive relations to both local and long-range synaptic integration deficits and support the diversity of tau and A β associated mechanisms of E/I imbalance”. This is overstated in terms of interpreting the physiological implications of what was measured.

We have revised the sentence to explicitly mention what we observed in the results as follows: “The differential associations in the current results— tau distinctly correlated with increased neural excitability and A β showing positive relations to both neural excitability and neural fragility indicates diversity of tau and A β associated mechanisms of AD pathophysiology relating to manifestations of E/I imbalance”. (This is addressed under the discussion section under the subheading “Epileptic manifestations signify an AD phenotype with greater neural fragility” starting on page 13)

- 15) In discussing the “epileptic phenotype of AD” - I presume that the AD-EPI+ participants mostly had subclinical epileptiform activity (based on prior publications from this group), but this is not well-described in the current paper. I think it is important to distinguish between AD patients with subclinical epileptiform activity and those with clear clinical epileptic seizures in the discussion. The underlying pathologic mechanisms may not necessarily be the same, and conflating the two in the discussion may undermine the authors’ point regarding the “diversity” of E/I imbalance that occurs in AD.

Our study explicitly reported subclinical epileptiform activity. This was an essential step in the careful clinical characterization of this cohort to exclude any confounds from primary epilepsy as a co-pathology. The detection of subclinical epileptiform events allowed us to conclude with high certainty that the epileptiform events observed in this cohort are likely associated with AD pathophysiology rather than primary epilepsy. We have clarified these details in the revised ‘Methods’. (page 20, line 7-11)

- 16) “The current finding that AD-EPI+ having higher neural fragility than AD-EPI-, despite similarly increased neural excitability, suggest that epileptic events are not a uniform manifestation but rather represent an AD phenotype with an added burden of E/I imbalance. As neural fragility is only correlated to A β , it is likely that this greater disease burden is contributed by A β vulnerable mechanisms—a phenomenon supported by both animal and clinical data.” The phrase “added burden of E/I imbalance” is confusing here, as the authors previously use E/I imbalance as a term that encompasses both neural excitability and neural fragility. Just because the measure of neural fragility was associated with AB, and neural fragility was higher in AD-EPI+, does not necessarily mean that AB is what drives the epileptic phenotype.

We agree with the reviewer’s point and have revised the sentence by removing the term “added burden.” Additionally, we have clarified the discussion regarding the relationship between amyloid and neural fragility, avoiding overstatement of our findings. (discussion section under the subheading “Epileptic manifestations signify an AD phenotype with greater neural fragility” starting on page 13)

- 17) The authors did not directly examine amyloid or tau pathology in the AD-EPI+ vs EPI- group comparisons. Why not? This should be stated as a major limitation.

We agree with the reviewer that quantification of amyloid and tau pathology in AD-EPI+ versus AD-EPI- patients represents an important next step to further elucidate the mechanistic associations between E/I imbalance and AD proteinopathy. However, amyloid- and tau-PET imaging data are not available for all patients in the current sub-cohorts defined by long-term video EEG and M/EEG protocols used to categorize AD-EPI+ and AD-EPI-. This discrepancy arises because the AD-EPI protocol, involving resource-intensive overnight video EEG monitoring followed by one-hour M/EEG, has been conducted over many years since 2013, whereas amyloid- and tau-PET imaging became available for research only more recently. Nonetheless, we have an ongoing longitudinal study employing multimodal imaging—including M/EEG, amyloid-PET, and tau-PET—in AD patients. This study will enable us to determine whether amyloid and tau pathology differ between AD-EPI+ and AD-EPI- groups and to relate E/I measures to protein deposition within each sub-cohort. We plan to publish these results in future manuscripts. We have included this limitation and future direction in the revised manuscript (please also see *Reviewer #1 - introduction: c*)(page 19, 'Limitations')

- 18) “Second, clinical studies in AD patients also suggest that seizures are more likely to be associated with amyloid pathology rather than tau pathology. Seizures and epileptic manifestations are more common in patients with early-onset AD and during the initial stages of the disease likely reflecting the association between increased seizure activity and higher A β burden.” I disagree with these statements and think a more balanced discussion is needed. If seizures in AD were more likely associated with amyloid pathology, then they would be seen more commonly in pre-clinical stages of AD (when amyloid burden is high, without significant tau accumulation). While prior studies have shown that seizures can occur in preclinical stages of AD, this does not happen in the majority of cases (Vossel et al, JAMA Neurology 2013; Sarkis et al, J Neuropsychiatry Clin Neurosci 2016), and many seizures happen at or after onset of clinical symptoms, which is more closely correlated with accumulation of tau pathology. Moreover, see DeVulder et al, Brain 2025, Lam et al, Neurology 2025, which demonstrate associations between local tau pathology and epileptiform activity and seizures in AD.

Thank you for this constructive criticism. Based on the suggestions from the reviewer we have now extensively revised the discussion section incorporating all the suggestions. The sentence that was pointed out here has been re-stated with clarity without overstating the findings. Furthermore, the important studies the reviewer pointed out have been added to relevant sections of the revised discussion. (page 14-15)

- 19) “The finding that A β pathology likely drives epileptic activity in the AD-EPI+ phenotype has important consequences for research focused on understanding how A β affects E/I imbalance.” The analyses done in this study do not lead to the conclusion that “A β pathology likely drives epileptic activity in the AD-EPI+ phenotype”. There was no analysis of amyloid or tau done in the AD-EPI+ group to directly support this statement.

In the revised discussion we have removed this sentence.

- 20) Section on “Implications on protein lowering therapeutic trials and novel targets in clinical trials” . I think this section should be shortened significantly. While I agree that these measures of E:I imbalance could potentially be interesting functional biomarkers to assess therapeutic effects in AD, it is not clear to me that these will necessarily be useful. Some of this discussion (regarding CAA and anti-seizure therapies) is not directly relevant to the findings of this paper.

Thank you for the criticism. We have revised and shortened this section of the discussion. (page 17, lines 6-20)

Limitations:

- 21) Should include that this is an early onset AD cohort, with mean age of 62, and even lower for the AD-EPI+ group (60) -- which means onset of clinical symptoms may have been in the late 50s, and it is not clear whether these results would hold in a more typical AD cohort.

We acknowledge this limitation in the revised manuscript. (page 19, 'Limitations')

Conclusions:

- 22) “Whereas tau distinctly affects local excitability processes, A β affects both local and long-range synaptic input integration processes, with a stronger influence on the latter.” Just because tau did not correlate with neural fragility, does not necessarily mean that it only affects local excitability processes. Neural fragility is just one measure of the larger network. “Importantly, AD-EPI+ phenotype represents an added burden of E/I imbalance specifically related to A β associated long-range integration deficits.” This conclusion is not totally clear to me.

We have addressed both these concerns in the revised conclusion. (revised conclusion starting on page 19)

Reviewer #3

Thank you.

Reviewer #4

This is an interesting study dealing with a topic of significant value in the attempt of understanding dementia pathophysiology, in particular for the Alzheimer’s Disease type. Methodology is advanced and sound enough and results/conclusions might be of general interest when reviewed under the comments/criticisms I will try to share with the Authors. A major comment/criticism is on the general topic, namely the changes in excitability of the AD brain and the linkage it might have with the increased rate of epilepsy with respect to non demented age-gender-education matched elderly population and with the neurodegenerative processes. Is this a specific marker of the demented brain? We know that in the ‘natural epidemiology’ of primary epilepsies there is a peak defined late onset epilepsy in elderly population. We also know that the risk of epilepsy is increased following an acute brain lesion like in stroke and in slowly growing lesion like for a brain neoplasm. What do have such conditions in common between them and AD? They have in common the acute/progressive loss of neurons, networks and connections and the brain reaction trying to resist to such a loss. One simple mechanism of brain resilience is the change in excitability in the attempt of recruiting ‘silent’ synapses/circuits to vicariate the function of the lost ones. In other words a simple and efficient mechanism which is active in several brain conditions from physiological brain aging to several and different brain pathologies; worth of mentioning, in all such conditions the rate of epilepsy is increased. On this basis how specific is what is described by the Authors within the frame of different ‘specific markers’ for AD? Moreover, there is a bulk of literature confirming what I just wrote utilizing Transcranial Magnetic Stimulation (TMS) and showing changes in brain excitability in the healthy elderly, in stroke, in AD and different forms of central nervous system neurodegeneration including amyotrophic lateral sclerosis. Authors are encouraged to consider such a comment and to include it in the ‘limitations’ section as well as in the introductory part of the text.

We thank the reviewer for this important suggestion. We completely agree with the reviewer’s perspective that E/I imbalance is a fundamental response of neural circuit dysfunction—where hyperexcitability can be either maladaptive or compensatory. Given its role as a core element in the neural circuit dysfunction quantifiable tools of E/I imbalance and their associations with the pathological markers of A β and tau, play critical in identifying the correct therapeutic targets in AD and evaluating the efficacy of interventions aimed at improving cognition. We have now added a new section into our revised discussion under the subheading “E/I imbalance as a core element of neural circuit dysfunction” clarifying this point. (page 17)

Other point –of less general impact on the experimental design- include:

- 1) Spiky EEG activity does not mean epilepsy since up to 10% of the general population can show them in the EEG without suffering any epileptic attack throughout their lives.. This aspect should be clarified. This observation tempers down the percentage of epilepsy of AD and dementias to about 10-15% of such population. Still higher than that physiological and healthy brain, but less than reported in the text.

We have now added these additional data points from healthy elderly into our revised discussion. (page 13, lines 20-22)

- 2) Mixing together early AD and MCI-due-AD might introduce a remarkable bias. It is in fact a common observation that in a MCI population only 30-to-50% progress to clinical dementia while the remaining is not even after a prolonged follow-up. This is also valid for MCI having positive biomarkers for beta/tau pathology even if at a lower level than those who have not. The reason for this is not clear, unless that the 'resilient' MCI amyloid/tau positive population must have protective factors which balance or block the activity of the risk factors. By putting together AD Patients and MCI subjects who might not develop dementia is introducing a confounding factor of importance in the final results.

We appreciate the comment and agree with the reviewer that the clinical syndrome of MCI represents a heterogenous group. In our study, all participants included as MCI (CDR=0.5) or mild AD (CDR=1), were tested for AD biomarkers and were confirmed positive which gives a high certainty for these individuals to be under the 'AD neuropathological spectrum'. Nonetheless, we have repeated all the LMMs examining the associations between E/I imbalance and AD proteinopathy as well as hypometabolism, including the categorical label of MCI (CDR=0.5) or mild AD (CDR=1) into the statistical models. We have included these results in the revised Supplementary materials. In brief, the avocations between E/I imbalance and AD proteinopathy as well as hypometabolism remained similar in CDR=0.5 and in CDR=1, and there was no significant interaction with CDR. We believe that this additional information will be helpful for the reader. (page 8, lines 17-22; Supplementary figure S1; page 10, lines 8-15; Supplementary figure S3)

- 3) The above point is strongly dealing with the debate between a 'biological' diagnosis and a 'clinical/neuropsychological diagnosis'. A consistent population of at-risk subjects (i.e. MCI population) is considered by the biological diagnosis supporters already affected by dementia when they carry positivity for given biomarkers even in absence of symptoms and progression and any significant impact in their living efficiency except for the tremendous burden due to a 'biological diagnosis' which will definitely change their social/professional/affective profile for several years until it is demonstrated that the 'biological diagnosis' was wrong in terms of a real disease.

Thank you for the comment. We completely agree with the reviewer that the concept of biological AD is a critical issue that needs a broader discussion in the field of AD research. Please also refer to the response above (#2).

REFERENCES

1. Sritharan D, Sarma SV. Fragility in dynamic networks: application to neural networks in the epileptic cortex. *Neural Comput* **26**, 2294-2327 (2014).
2. Fan L, *et al.* The Human Brainnetome Atlas: A New Brain Atlas Based on Connectional Architecture. *Cereb Cortex* **26**, 3508-3526 (2016).
3. Ranasinghe KG, *et al.* Neuronal synchrony abnormalities associated with subclinical epileptiform activity in early-onset Alzheimer's disease. *Brain* **145**, 744-753 (2022).
4. Ranasinghe KG, *et al.* Neurophysiological signatures in Alzheimer's disease are distinctly associated with TAU, amyloid-beta accumulation, and cognitive decline. *Sci Transl Med* **12**, (2020).
5. Hinkley LB, Vinogradov S, Guggisberg AG, Fisher M, Findlay AM, Nagarajan SS. Clinical symptoms and alpha band resting-state functional connectivity imaging in patients with schizophrenia: implications for novel approaches to treatment. *Biol Psychiatry* **70**, 1134-1142 (2011).

6. Stam CJ, *et al.* Graph theoretical analysis of magnetoencephalographic functional connectivity in Alzheimer's disease. *Brain : a journal of neurology* **132**, 213-224 (2009).
7. Cuesta P, *et al.* Source analysis of spontaneous magnetoencephalographic activity in healthy aging and mild cognitive impairment: influence of apolipoprotein E polymorphism. *J Alzheimers Dis* **43**, 259-273 (2015).
8. Bosma I, *et al.* The influence of low-grade glioma on resting state oscillatory brain activity: a magnetoencephalography study. *J Neurooncol* **88**, 77-85 (2008).
9. Jia Y, Jariwala N, Hinkley LBN, Nagarajan S, Subramaniam K. Abnormal resting-state functional connectivity underlies cognitive and clinical symptoms in patients with schizophrenia. *Front Hum Neurosci* **17**, 1077923 (2023).
10. Hari R, *et al.* IFCN-endorsed practical guidelines for clinical magnetoencephalography (MEG). *Clin Neurophysiol* **129**, 1720-1747 (2018).
11. Ehrens D, Li A, Aeed F, Schiller Y, Sarma SV. Network Fragility for Seizure Genesis in an Acute in vivo Model of Epilepsy. *Annu Int Conf IEEE Eng Med Biol Soc* **2020**, 3695-3698 (2020).
12. Li A, *et al.* Neural fragility as an EEG marker of the seizure onset zone. *Nat Neurosci* **24**, 1465-1474 (2021).
13. Donoghue T, *et al.* Parameterizing neural power spectra into periodic and aperiodic components. *Nat Neurosci* **23**, 1655-1665 (2020).
14. Lendner JD, *et al.* Human REM sleep recalibrates neural activity in support of memory formation. *Sci Adv* **9**, eadj1895 (2023).
15. Brake N, *et al.* A neurophysiological basis for aperiodic EEG and the background spectral trend. *Nat Commun* **15**, 1514 (2024).
16. Colombo MA, *et al.* The spectral exponent of the resting EEG indexes the presence of consciousness during unresponsiveness induced by propofol, xenon, and ketamine. *Neuroimage* **189**, 631-644 (2019).
17. Ahmad J, *et al.* From mechanisms to markers: novel noninvasive EEG proxy markers of the neural excitation and inhibition system in humans. *Transl Psychiatry* **12**, 467 (2022).

Review Report of the manuscript “Distinct manifestations of excitatory-inhibitory imbalance associated with amyloid- β and tau in patients with Alzheimer’s disease” for publication in nature communications.

Ranasinghe and colleagues are reporting on magnetoencephalography (MEG) differences between Alzheimer’s disease patients and controls and the association with 1) proteinopathies (tau and amyloid) and neurodegeneration and 2) epileptiform activity. The study claims that their approach helps to understand the how Excitation/Inhibition (E/I) balance is altered across the brain and its distinct associations with amyloid, tau and neurodegeneration, and whether patients with epileptiform activity present a distinct group of patients.

In general, the need for better understanding the role of E/I imbalance in AD patients and its relation with amyloid, tau and neurodegeneration in human AD is high. The use of MEG and PET scanning to link large-scale neuronal function to underlying proteinopathies is a promising approach, and the author group possesses substantial expertise in this field. The methods used are innovative and a subset of the cohort consists of unique multimodal imaging data, including Amyloid, tau, FDG-PET and MEG. The finding that 1) amyloid and tau present distinct associations with E/I balance and 2) AD patients with epileptiform activity present a distinct group of patients are highly relevant for future therapeutic implications. The structure, clarity and context of the manuscript is of high quality. The sample size of this study appears adequate, the cohorts are well-characterized, and the conclusions are supported by results. I therefore consider the aim and the scope of this paper suitable for publication in Nature Communications. I only would like to share minor remarks which I would like to ask the authors to address before being able to recommend the manuscript for publication.

1) General remarks

- a. Overall, local excitability and interregional synaptic integration are presented as two distinct, complementary mechanisms throughout the paper. This is also supported by a lack of correlation between the measures in control subjects and the distinct spatial distributions over the cortex. However, local excitability is also greatly dependent on synaptic inputs from other regions, and, therefore, I think one should be careful in presenting local excitability and interregional synaptic integration as two distinct entities. Please, could the authors reflect upon this matter and, if they agree, nuance these statements throughout the manuscript?
- b. Furthermore, the authors correctly chose the aperiodic activity as a measure of local neuronal excitability. However, it would be fair to introduce that various potential quantitative measures of E/I exist for M/EEG [for example, see Ahmad et al. 2022 Transl Psych]. Most of these (including 1/f [A systematic review of aperiodic neural activity in clinical investigations | medRxiv]) are at their infancy, requiring further research to elaborate upon their robustness and validity. Furthermore, the use of 1/f as measure of excitability is not straightforward [Brake et al 2023 Nat Comm], which is supported by the findings that 1/f does not associated with epileptiform activity in the current study. The limitations of the chosen measures to quantify E/I should at least be addressed in the discussion section.

2) Introduction

- The authors conclude in the first paragraph that none of the AD mouse models capture the full complexity of AD dual proteinopathy, which I think is true. However, there have been several approaches aiming to model both amyloid and tau pathology in mice, which deserve some attention (for instance: Tok, Crauwels and Drinkenburg (2022) Journal of Neurology and Experimental Neuroscience.).

- I have some conceptual remarks about the use of the words vulnerability and heterogeneity.
 - o The authors state that AD patients with epileptiform activity present ‘additional disease vulnerabilities’ and hypothesize that these subjects have ‘greater vulnerabilities of E/I balance’, but it is not clear to me what is meant with vulnerability here. Does it mean that the same load of amyloid or tau pathology elicits more severe E/I imbalance (resulting in epileptiform abnormalities) compared to other patients? If so, what would be potential mechanisms for this increased vulnerability? Or does it mean that similar E/I values can have distinct manifestations (if so why)? Please, could the authors elaborate upon this?
 - o One of the provided explanations for why only a subset of patients show epileptiform activity is “heterogeneity”. The authors subsequently explain a potential differential (i.e. heterogeneous) effect of amyloid and tau on E/I balance. Is this indeed the type of heterogeneity meant to explain differential presentation of E/I imbalance among AD patients or do the authors refer to a different kind of heterogeneity? (this also accounts for use of the word heterogeneity in discussion section). If the former is true, would it make more sense to compare the load of abeta and tau between EPI+ and EPI- subjects?

3) Methods

- There is some missing information:
 - o A statement that the study has been performed in accordance with the declaration of Helsinki
 - o The atlas used for source reconstruction of MEG data, and why the authors included only 210 cortical out of total of 246 ROIs
 - o The authors mention in the results section that for some statistical models CDR was included as covariate, but this is not reported in the methods, please check.
 - o Did the authors check assumptions for LMM (such as normal distribution, constant variance)? Did the authors expect a linear relationship between the measures?
- A number of methodological choices require some additional argumentation or explanation:
 - o Could the authors explain why for tau and amyloid the SUVR values are computed, but FDG PET is normalized to control data?
 - o Why are z-scores of E/I measures computed and used for some, but not all, statistical analyses?
 - o What is an age-corrected unpaired t-test (line 6 p 23)? Do the authors mean ANCOVA?
- Some other thoughts:
 - o Should CDR or another indicator of ‘disease stage/duration’ be considered as covariate for all models (do the authors expect similar correlations for MCI vs probable AD stage?)
 - o Why did the authors chose to estimate the association between E/I and neurodegeneration after correcting for abeta/tau and not, for example, estimating the association between E/I and tau when correcting for amyloid? I think the latter would have given additional evidence about the specificity of tau with E/I manifestations, independent of amyloid.
 - o One may expect that the locations showing distinct fragility will overlap with epileptiform activity location, but this is not reported upon. I believe this data should be available from previous reports and including this in the manuscript would enhance the credibility of the findings.

4) Discussion

- It seems as if the authors tend to infer a direction of effect between E/I measures and PET (“E/I imbalance is driven by/stemming from .. PET”), but considering the cross-sectional nature of the data, one should be careful making such statements. I would suggest to nuance these statements a bit to avoid over interpretation of results.
- The authors report distinct neural fragility between EPI+ and EPI- AD patients in specific locations, but similar local neural excitability. Do these findings suggest 1/f is not an appropriate measure for E/I?
- In general, I believe the discussion would benefit from inclusion of a short paragraph comparing results to previous clinical MEG reports and combined MEG/PET studies in AD, since this is not the first paper studying the relation between E/I in AD and controls or MEG in relation to abeta/tau. [Wiesman et al 2022 Brain; van Nifterick et al 2023 SciRep; Martinez-Canada et al 2023; Gallego-Rudolf et al 2024 NatComm; Schoonhoven et al., 2023 Brain; Javed et al 2025 JoN]. Also, a key finding of this study is a strong correlation between local hyperexcitability and neurodegeneration; How does this relate to previous literature?
- Please, provide some words on the limitations of used metrics for E/I in the discussion section (as mentioned before)